# MMBench-Video: A Long-Form Multi-Shot Benchmark for Holistic Video Understanding

**Xinyu Fang**[1,2*], **Kangrui Mao**[3*], **Haodong Duan**[2*†],
**Xiangyu Zhao**[2,3], **Yining Li**[2], **Dahua Lin**[2,4,5], **Kai Chen**[2†]
[1]Zhejiang University     [2]Shanghai AI Laboratory     [3]Shanghai Jiao Tong University
[4]The Chinese University of Hong Kong     [5]CPII under InnoHK
[*] Equal Contribution     [†] Corresponding Author
fangxinyu,duanhaodong@pjlab.org.cn

## Abstract

The advent of large vision-language models (LVLMs) has spurred research into their applications in multi-modal contexts, particularly in video understanding. Traditional VideoQA benchmarks, despite providing quantitative metrics, often fail to encompass the full spectrum of video content and inadequately assess models' temporal comprehension. To address these limitations, we introduce MMBench-Video, a quantitative benchmark designed to rigorously evaluate LVLMs' proficiency in video understanding. MMBench-Video incorporates lengthy videos from YouTube and employs free-form questions, mirroring practical use cases. The benchmark is meticulously crafted to probe the models' temporal reasoning skills, with all questions human-annotated according to a carefully constructed ability taxonomy. We employ GPT-4 for automated assessment, demonstrating superior accuracy and robustness over earlier LLM-based evaluations. Utilizing MMBench-Video, we have conducted comprehensive evaluations that include both proprietary and open-source LVLMs for images and videos. MMBench-Video stands as a valuable resource for the research community, facilitating improved evaluation of LVLMs and catalyzing progress in the field of video understanding. The evaluation code of MMBench-Video will be integrated into VLMEvalKit: https://github.com/open-compass/VLMEvalKit.

## 1   Introduction

As a ubiquitous format for multimedia, video holds a pivotal role in people's lives, serving purposes such as knowledge dissemination, sharing life experiences, and entertainment. The rapid proliferation of video content has reshaped communication, learning, and connection in the digital age. The vast amount of online video content underscores the importance of algorithmic video understanding. Current video understanding paradigms, which often focus on specific tasks [20, 66, 19], typically excel only on in-domain data. An ideal video understanding system should demonstrate robust zero-shot capabilities, accurately discern contextual, emotional, and linguistic details within a video, and engage in free-form dialogues with humans [31, 39].

With the rapid development of Large Language Models (LLMs) [41, 52, 53], Large Vision Language Models (LVLMs) [42, 51, 37, 17] have also seen significant advancements. Typical video-language models developed by researchers utilize frame-level [39] or clip-level [34] visual features extracted by vision encoders [47, 55, 23], align these features with language embeddings via a projector, and process these embeddings with a fine-tuned large language encoder [14]. The models are fine-tuned with video instruction data and quantitatively assessed on free-form VideoQA benchmarks [62, 25, 56]. The current evaluation of Video-LLMs is characterized by the following limitations:

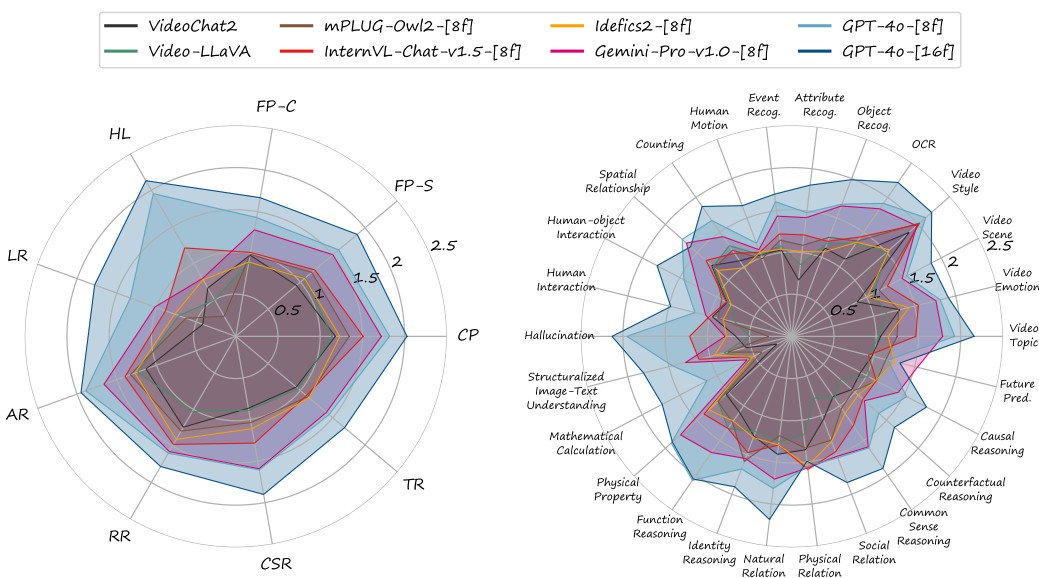

Figure 1: **Comparing mainstream LVLMs on MMBench-Video.** Two radar graphs illustrate the performance for each coarse (L-2) and each fine-grained (L-3) capability, respectively.

1. **Short Videos**: Existing VideoQA datasets primarily consist of short videos, typically lasting less than a minute. Meanwhile, most web video content spans several minutes or longer, creating a discrepancy between the evaluation benchmark and real-world application scenarios.

2. **Limited Capabilities**: Current VideoQA benchmarks are limited to several basic video tasks [24], including concept existence, object relationship recognition, and activity recognition. There are more fine-grained perception and reasoning capabilities [38] not encompassed by existing benchmarks.

3. **Biased Evaluation**: Existing evaluation paradigms employ GPT-3.5 to score open-ended answers generated by video-language models. Our preliminary study indicates that GPT-3.5-based evaluation is less accurate and exhibits significant discrepancy relative to human preferences, diminishing the credibility of the evaluation results.

To address these problems, we develop a new VideoQA benchmark, **MMBench-Video**, to evaluate the effectiveness of LVLMs in video understanding. It incorporates approximately 600 web videos with rich context from YouTube, spanning 16 major categories, including News, Sports, *etc.*, covering most video topics people watch in their daily lives. Each video ranges in duration from 30 secs to 6 mins, to accommodate the evaluation of video understanding capabilities on longer videos. The benchmark includes roughly 2,000 original question-answer (QA) pairs, contributed by volunteers, covering a total of 26 fine-grained capabilities. During dataset collection, we implement quality control strategies to explicitly increase the proportion of temporal indispensable questions[1]. Quantitative statistics show that MMBench-Video significantly differs from existing benchmarks in terms of temporal duration, context richness, and temporal indispensability.

During evaluation, an LVLM produces free-form responses to visual questions. Given the variability in the lengths and styles of ground-truth answers, accurately assessing these responses presents a significant challenge. In light of the limitations observed in previous evaluations powered by GPT-3.5, we propose the use of the more powerful GPT-4 [42] for automated scoring. This approach prioritizes semantic similarity while overlooking minor discrepancies in language organization. Employing a carefully crafted evaluation prompt, our GPT-4-based evaluation exhibits improved quality in terms of accuracy, consistency, and alignment with human judgment.

Based on MMBench-Video, we perform a thorough evaluation of mainstream LVLMs, including open-source video-language models (Video-LLMs), as well as both open-source and proprietary LVLMs for image understanding. We report their performance across diverse capabilities, as depicted in Fig. 1. The performance rankings enable direct comparisons between models, revealing critical insights into

---

[1]A visual question is temporal indispensable if it can not be correctly solved by viewing any random frame.

their limitations. Surprisingly, existing Video-LLMs exhibit subpar performance on MMBench-Video, significantly underperforming proprietary LVLMs and even lagging behind open-source LVLMs, such as Idefics2 [28] and InternVL-Chat-v1.5 [13]. To further investigate these models' capabilities, we employ image VQA benchmarks to assess their image understanding skills, again observing a substantial gap between Video-LLMs and the state-of-the-art LVLMs. The comprehensive assessment underscores the significant performance disparities between Video-LLMs and leading LVLMs in both spatial and temporal understanding, highlighting areas requiring future improvement.

In summary, the contributions of this work are as follows:

- **Innovative VideoQA Benchmark:** MMBench-Video features long-form, diverse videos sourced from the web, encompassing a broad spectrum of topics. It includes original, high-quality visual questions crafted by volunteers, spanning dozens of fine-grained capabilities.
- **Enhanced Scoring Methodology:** We assess the limitations of using low-quality LLMs, such as GPT-3.5, for scoring model responses. To address this, we implement a GPT-4-based evaluation paradigm, which offers superior accuracy, consistency, and a closer alignment with human judgments.
- **In-depth Evaluation:** Our comprehensive assessment of various LVLMs on MMBench-Video reveals detailed insights into their performance across multiple fine-grained capabilities. The results underscore the current limitations of Video-LLMs in spatial and temporal understanding, guiding future research and development.

## 2 Related Work

### 2.1 Large Vision-Language Models

The success of Large Language Models (LLMs) such as GPTs [47, 7, 42] and LLaMA [52, 53] has spurred significant advancements in Large Vision-Language Models (LVLMs). Flamingo [3] has demonstrated impressive few-shot capabilities by integrating gated cross-attention blocks to connect pre-trained vision and language models. BLIP [30, 17] employs a Querying Transformer to bridge the modality gap between a frozen image encoder and a language encoder. LLaVA [37] leverages GPT-4 to create instruction-following data for vision-language tuning, with its learning paradigm and instruction tuning corpus being widely adopted by subsequent works [36, 11, 1, 16]. In the realm of video-language models, Video-ChatGPT [39] aligns frame-level vision features with language embeddings via a linear projector, whereas VideoChat [31] utilizes a learnable Q-former, inspired by BLIP-2. Subsequent works like Video-LLaMA [63] integrate audio features, and Video-LLaVA [34] learns from a mixed dataset of images and videos. Additionally, proprietary APIs such as GPT-[4v/4o] [42], Gemini [51], and Reka [45] have been made publicly available, supporting various input formats including single or multiple images. We present a comprehensive evaluation of existing LVLMs, encompassing Video-LLMs as well as open-source and proprietary LVLMs for images, using the proposed MMBench-Video to provide a detailed landscape of their capabilities.

### 2.2 Video Question Answering

Video Question Answering (VideoQA) is a critical method for assessing the depth of understanding that models possess regarding video content. The research community has progressively developed a wide array of VideoQA benchmarks, spanning various visual domains such as movies [50], TV shows [29, 22], video games [40], synthetic scenarios [61], and egocentric videos [21]. These benchmarks typically assess models trained on their respective training sets, demanding concise answers for evaluation. However, Large Vision and Language Models (LVLMs), which are often not trained on domain-specific data, face challenges in adapting to these benchmarks due to their diverse answer styles. To mitigate this, Video-ChatGPT [39] employs GPT-3.5 as a scoring mechanism for free-form responses from VLMs. The method was applied to evaluate several popular benchmarks [62, 25, 56], covering topics including concept existence, objection relationship, and activity recognition. Despite its broad adoption [34, 58, 48, 65], this approach is limited by suboptimal accuracy and stability, as well as poor alignment with human preferences. Additionally, those benchmarks primarily consist of short videos, which contrasts with the typical length of web videos. In response, we present MMBench-Video, a novel dataset tailored for longer videos, challenging models to generate detailed, free-form responses to complex questions. We adopt GPT-4-based evaluation, which improves correctness and robustness, offering a more stable evaluation strategy compared to previous methods.

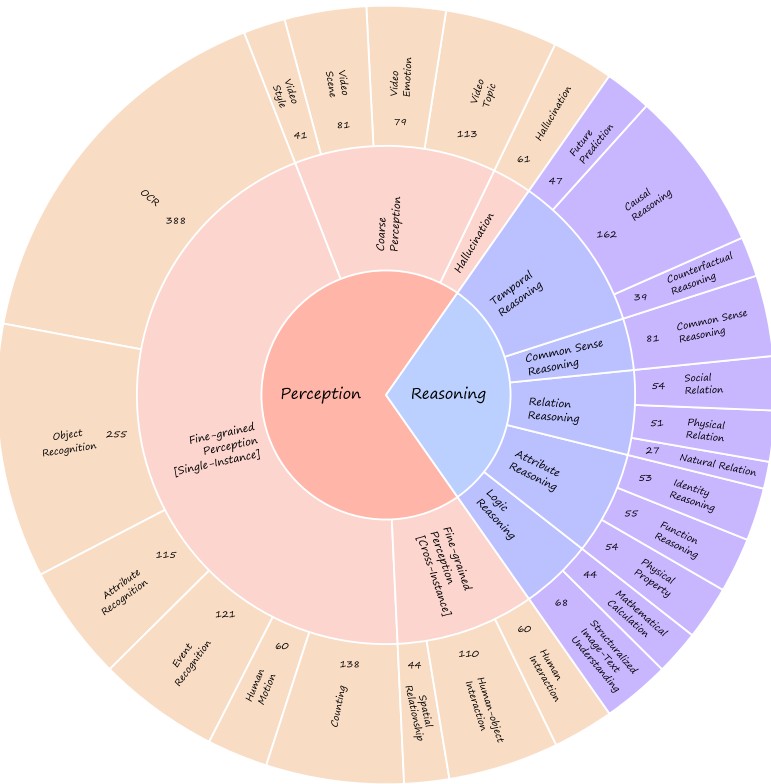

Figure 2: **Overview of ability dimensions in MMBench-Video.** Currently, MMBench-Video incorporates three levels of ability dimensions (L-1 to L-3), encompassing 26 distinct leaf abilities.

## 3 MMBench-Video

In this section, we delve into the meticulous construction of MMBench-Video, outlining our strategic approach to video question selection, the conceptualization and design of a comprehensive capability taxonomy, and the innovative methods employed to enhance the temporal relevance and quality of questions. Additionally, we present detailed statistics of MMBench-Video and contrast it with existing VideoQA benchmarks, thereby illustrating its unique features and contributions to the field.

### 3.1 Benchmark Construction

**Video Collection.** To create a VideoQA benchmark, a prevalent approach involves generating question-answer pairs for videos sourced from existing datasets. For example, MSRVTT-QA and MSVD-QA are derived from video retrieval datasets [57, 10], while ActivityNet-QA is constructed using an action recognition dataset [8]. Most existing VideoQA datasets are limited to short videos with a constrained number of shots, exhibiting limited diversity in content. To develop a benchmark that more closely mirrors the web video content commonly consumed by viewers, we propose the creation of a long-form, multi-shot VideoQA benchmark. The benchmark draws its content directly from YouTube, offering several distinct advantages. Firstly, YouTube's extensive metadata, including video titles, click metrics, and subtitles, provides valuable context for video understanding. Secondly, as a leading global streaming platform, YouTube's vast user base ensures the dataset's diversity.

Drawing inspiration from the YouTube-8M [2] labels, our categorization scheme encompasses 16 major categories (Fig. 3), spanning from engaging topics like 'Entertainment and Sports' to enlightening subjects such as 'Science and Knowledge'. Volunteers are instructed to navigate through YouTube and collect videos that align with these designated categories. In line with our objective to amass long-form content, volunteers are directed to disregard videos with durations of less than 30 seconds. Although we impose no upper limit on the length of the web videos collected, all

Table 1: **Comparing the statistics of MMBench-Video and other widely adopted VideoQA benchmarks.** When reporting the video statistics, we follow the format of "mean value (standard deviation)".

| Benchmarks | QA pairs Generation | Number of Capabilities | Question Length mean(std) words | Answer Length mean(std) words | Video Duration mean(std) sec | Shot Number mean(std) |
|---|---|---|---|---|---|---|
| MSVD-QA [56] | Automatic | 2 | 6.6(2.5) | 1.0(0.0) | 9.8(6.6) | 2.4(3.4) |
| MSRVTT-QA [57] | Automatic | 2 | 7.4(3.4) | 1.0(0.0) | 15.1(5.2) | 3.4(2.9) |
| TGIF-QA [25] | Automatic/Human | 4 | 9.7(2.3) | 1.5(0.9) | 3.7(2.0) | 1.2(1.4) |
| ActivityNet-QA [62] | Human | 3 | 8.9(2.4) | 1.3(0.7) | 111.5(66.1) | 12.9(20.9) |
| MMBench-Video | Human | **26** | **10.9**(4.1) | **8.4**(7.7) | **165.4**(80.7) | **32.6**(33.5) |

question-answer pairs composed for a video will be derived from a clip no longer than 6 minutes. This helps maintain a practical balance between video duration and the task complexity.

**Capability Taxonomy.** Inspired by MMBench [38], we have developed a 3-level (L-1 to L-3) hierarchical capability taxonomy (Fig. 2). The top level encompasses two broad capabilities: Perception and Reasoning. Besides the six L-2 capabilities inherited from MMBench, we further introduce three additional L-2 capabilities specific to MMBench-Video: Hallucination, Commonsense Reasoning, and Temporal Reasoning. **Hallucination** assesses whether a model is prone to generating content that includes misleading or inaccurate information. **Commonsense Reasoning** evaluates a model's ability to integrate necessary commonsense knowledge into its reasoning processes. **Temporal Reasoning** examines a model's proficiency in understanding the relationships between events unfolding at different video points. This taxonomy comprises a total of 26 leaf capabilities, which collectively address a comprehensive spectrum of cognitive processes involved in video comprehension.

**Composing Questions and Answers.** A well-known issue in existing VideoQA benchmarks is the prevalence of non-temporal questions, which are those that can be accurately answered based on nearly any frame within a video, rendering them effectively 'static'. These questions fail to adequately assess a model's ability to temporal understanding. In the curation of MMBench-Video, we prioritize questions that necessitate temporal reasoning and strive to minimize the occurrence of static questions. Recognizing the necessity of evaluating certain coarse perception capabilities, such as Video Style and Video Topic, it is impractical to entirely eliminate static questions. Instead, we focus on significantly reducing their proportion within the benchmark.

In MMBench-Video, each video is accompanied by multiple independent questions designed to assess one or more specific leaf capabilities. For instance, a question that requires identifying and counting a particular type of object would evaluate both Object Recognition and Counting capabilities. To ensure the quality and relevance of the questions and their corresponding answers, volunteers involved in the collection process are provided with the following five guidelines to adhere to:

1. Each question should **evaluate one or multiple leaf capabilities** within the established taxonomy.
2. You are encouraged to formulate **temporal indispensable questions**, as long as it's feasible for the corresponding video content and capability category.
3. **Avoid including specific timestamps** in the questions, such as "at 03:20 in the video". Please use relative expressions like "at the end of the video" or "before/after a specific event" instead.
4. The questions should be **free-form** and exhibit **linguistic diversified**. Besides standard formats like *What/Who/How*, questions can also adopt a conversational style[2].
5. Please provide **informative and detailed answers** for each question.

All generated question-answer pairs in MMBench-Video will be subjected to a meticulous cross-validation process to confirm their accuracy and adherence to the established guidelines. In addition to this, we implement an LVLM-based filtering mechanism to identify and eliminate a portion of static questions, as detailed in the supplementary material. The final MMBench-Video dataset comprises a diverse selection of web videos sourced from YouTube, accompanied by human-composed, original question-answer pairs designed to assess a comprehensive array of fine-grained capabilities.

**Evaluation Paradigm.** Given the varied length and style of ground-truth answers, automated robust evaluation that aligns with human judgments can be challenging. To address this, we propose a 3-grade marking scheme and utilize GPT-4 [42] as our adjudicator. GPT-4 assigns a score from 0 to 3 based

---

[2]For example, "What is the score of the football game in the video?" (a "what" question) can be expressed as "Tell me the winning team and the final score." (conversation style).

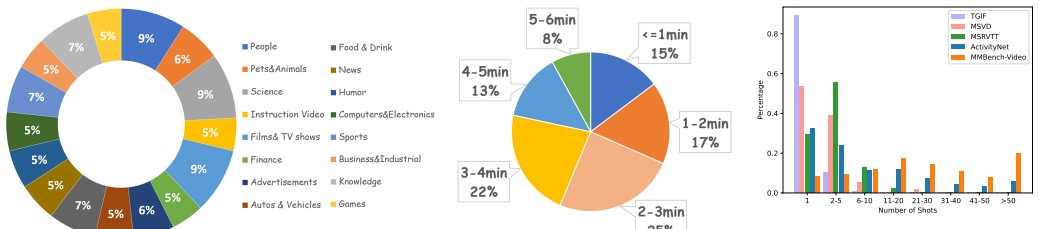

Figure 3: **Video category distribution of MMBench-Video.**

Figure 4: **Duration distribution of MMBench-Video.**

Figure 5: **Shot number distribution of MMBench-Video and other benchmarks.**

| Benchmark | MSVD | | TGIF | | MSRVTT | | ActivityNet | |
|---|---|---|---|---|---|---|---|---|
| Input Frames | 1 | 8 | 1 | 8 | 1 | 8 | 1 | 8 |
| Original Score | 2.62 | 2.93 | 2.66 | 3.18 | 2.01 | 2.33 | 2.65 | 3.05 |
| Normalized Score | 52.4 | 58.6 | 53.2 | 63.6 | 40.2 | 46.6 | 53.0 | 61.0 |
| Score-[1f] / Score-[8f] | 89.4% | | 80.5% | | 86.3% | | 87.0% | |
| Benchmark | EgoSchema | | Video-MME* | | Next-GQA | | MMBench-Video | |
| Input Frames | 1 | 8 | 1 | 8 | 1 | 8 | 1 | 8 |
| Original Score | 0.65 | 0.70 | 0.54 | 0.68 | 0.78 | 0.84 | 0.78 | 1.63 |
| Normalized Score | 65.0 | 70.0 | 54.0 | 68.0 | 78.0 | 84.0 | **26.0** | 54.3 |
| Score-[1f] / Score-[8f] | 88.6% | | 79.4% | | 92.9% | | **47.8%** | |

Table 2: **Comparing the temporal indispensability of existing VideoQA benchmarks.** MMBench-Video adopts a different grading paradigm compared to other benchmarks (3-grade *vs.* 5-grade). We calculate the 'Normalized Score' (normalize to 0-100) to ease the cross-benchmark comparisons. Benchmarks with smaller 1-frame scores and Score-[1f] / Score-[8f] ratio feature better temporal indispensability. For Video-MME, we excluded long-duration videos in this comparison to better align with other datasets.

on the content similarity between the model's output and the ground truth. Our experiments show that this evaluation framework exhibits strong consistency and alignment with human assessments.

## 3.2 Dataset Statistics

MMBench-Video comprises **609** video clips across 16 major categories, as depicted in Fig. 3), with durations spanning from 30 seconds to 6 minutes. The dataset has an average video length of **165 seconds**, totaling **28 hours** in aggregated duration. The duration distribution of the clips within MMBench-Video is illustrated in Fig. 4. The dataset includes **1,998** question-answer (QA) pairs, with each QA assessing one or multiple capabilities of a vision-language model. The distribution of QAs corresponding to each capability is visualized in Fig. 2. To highlight the distinct value of MMBench-Video, we compare its statistics with those of existing VideoQA benchmarks:

**Duration & Shot Numbers**[3]. MMBench-Video is specifically designed as a **long-form, multi-shot** video dataset. As indicated in Tab. 1, our dataset boasts a substantially greater average duration than existing benchmarks. As shown in Fig. 5, videos in our benchmark display a long-tail distribution in shot numbers, with a maximum of 210 shots. This significantly surpasses all other benchmarks in average shot count.

**Linguistic Characteristics of QAs.** MMBench-Video features free-form video QA with rich linguistic diversity. In benchmarks such as MSVD-QA and MSRVTT-QA, questions are automatically generated and invariably begin with pronouns such as 'what', 'who', *etc*. Conversely, a significant proportion of questions in MMBench-Video are framed in a conversational manner, enhancing linguistic diversity (further details are available in the supplementary materials). Regarding answers, previous VideoQA benchmarks often provide responses that are limited to a single word or a brief phrase. In contrast, MMBench-Video strives to offer more comprehensive answers that extend beyond a single word. This is evident in the distribution of answer lengths, as shown in Tab. 1.

**Capability Coverage.** Existing benchmarks typically cover only a limited set of fine-grained capabilities [24] and often lack an explicit capability taxonomy. For instance, the majority of questions in MSVD-QA and MSRVTT-QA assess the ability to determine the existence of concepts

---

[3]We adopt the open-source tools scenedetect [9] to obtain the shot number of a video.

(such as humans or objects) and to recognize relationships between objects. In contrast, ActivityNet-QA and TGIF-QA extend this by including assessments of activity recognition and repetition counting. In MMBench-Video, we have established a comprehensive taxonomy encompassing 26 fine-grained capabilities, with each capability being evaluated using dozens to hundreds of original QAs.

**Temporal Indispensability.** In contrast to existing VideoQA benchmarks, MMBench-Video is designed to be temporal indispensable. In a preliminary study, we find that a great proportion of QAs in existing datasets can be correctly answered by LVLMs without providing the temporal context. The underlying factors can be categorized into two primary ones: (1) The brevity of source videos, characterized by the limited number of shots, allows for its content to be adequately represented by a single frame. (2) Many of the QAs are too simplistic and can be answered through guesswork rather than comprehension. For instance, MSVD-QA and MSRVTT-QA are replete with 'who' questions, which are commonly answered with general terms like 'someone', 'man', or 'woman'. In MMBench-Video, we have made significant efforts to mitigate these factors.

To quantitatively measure the temporal indispensability of each VideoQA benchmark, we randomly sample 1000 QAs from orginal VideoQA datasets and comprehensive video understanding benchmark, and conduct a study on the subsets as well as MMBench-Video. We evaluate GPT-4o (by far the most powerful LVLM) on these benchmarks under 1-frame and 8-frame settings, and present the results in Tab. 2. Notably, GPT-4o using a 1-frame input achieves a normalized score of approximately 50%, retaining over 75% of its performance compared to an 8-frame input across previous VideoQA datasets. Even the latest benchmarks for comprehensive video understanding struggle to fully assess a model's temporal capabilities. In contrast, when assessed on the MMBench-Video, GPT-4o using a 1-frame input preserves only 47.8% of its efficacy compared to its performance with an 8-frame input, yielding a normalized score of just 26.0%. This marked difference underscores the temporal importance of MMBench-Video.

## 4 Experiment

Utilizing MMBench-Video, we assess a diverse array of large vision-language models (LVLMs), encompassing Video-LLMs and image-based LVLMs, both open-source and proprietary. For Video-LLMs, we utilize the default hyperparameters specified in their respective open-source implementations for inference. For image-based LVLMs, we conduct evaluations based on VLMEvalKit [15], employ greedy decoding during inference and cap the maximum number of output tokens at 512.

### 4.1 Main Results

**Open-Source Video-LLMs.** We first identify and evaluate representative open-source Video-LLMs using MMBench-Video. Adhering to their default settings, these Video-LLMs process a sequence of video frames, with the number of frames varying from eight to dozens. Interestingly, we observe that all Video-LLMs exhibit comparably subpar performance on MMBench-Video, despite notable performance disparities on other benchmarks. For instance, VideoChat2 surpasses Video-ChatGPT by 18% on the MSVD-QA score (3.9 vs. 3.3), yet the performance gap narrows to just 6% on the MMBench-Video score (0.99 vs. 0.93). All video LLMs attain an average score close to 1 (out of a total of 3), with the top-performing model LLaVA-NeXT-Video [64] reaching a mere 1.14. These findings suggest that the current state of video models' proficiency in understanding MMBench-Video is nascent, underscoring the challenges and emphasizing the necessity for advancements in video LLMs to enhance their capability and effectiveness in interpreting varied video content.

**Open-Source LVLMs for Images.** A significant number of LVLMs [18, 37, 6, 59, 28, 13, 44, 54] have been developed to comprehend image content and execute visual reasoning tasks. During our evaluation, we focused on LVLMs that support the multi-image inference interface. We assess several prominent open-source LVLMs: Idefics2-8B [28], Qwen-VL-Chat [6], mPLUG-Owl2 [60], InternVL-Chat-v1.5 [13], InternVL2 [13] and VILA1.5 [35] using MMBench-Video. To ascertain the models' ability to effectively leverage multiple input frames, we evaluated them under two distinct settings: 1-frame and 8-frame inputs. Results in Tab. 3 indicate that all models, except Qwen-VL-Chat, exhibit a substantial enhancement in performance when utilizing 8 frames compared to a single frame. Notably, VILA-1.5-40B emerges as the top performer, achieving an impressive average score of 1.61 with 14 frames as inputs, significantly surpassing all other evaluated video LLMs.

| Model | Overall Mean | Perception | | | | | Reasoning | | | | | |
|---|---|---|---|---|---|---|---|---|---|---|---|---|
| | | CP | FP-S | FP-C | HL | Mean | LR | AR | RR | CSR | TR | Mean |
| *LLMs* | | | | | | | | | | | | |
| GPT-4o [43] | 0.25 | 0.03 | 0.11 | 0.07 | 1.82 | 0.16 | 0.39 | 0.55 | 0.32 | 0.30 | 0.55 | 0.45 |
| *Open-Source Video-LLMs* | | | | | | | | | | | | |
| Video-ChatGPT-[100f] [39] | 0.93 | 0.91 | 0.94 | 0.81 | 0.39 | 0.90 | 0.70 | 1.15 | 1.12 | 0.84 | 0.94 | 0.97 |
| Video-LLaVA-[8f] [34] | 1.05 | 1.14 | 1.08 | 0.88 | 0.50 | 1.04 | 0.72 | 1.23 | 1.03 | 0.89 | 0.97 | 0.99 |
| Chat-UniVi-[64f] [26] | 0.99 | 1.07 | 1.00 | 0.93 | 0.39 | 0.98 | 0.59 | 1.18 | 1.14 | 0.75 | 0.98 | 0.97 |
| LLaMA-VID-[1fps] [33] | 1.08 | 1.30 | 1.09 | 0.93 | 0.42 | 1.09 | 0.71 | 1.21 | 1.08 | 0.83 | 1.04 | 1.02 |
| VideoChat2-[16f] [32] | 0.99 | 1.18 | 0.94 | 0.98 | 0.66 | 0.98 | 0.42 | 1.13 | 1.24 | 0.86 | 0.94 | 0.95 |
| MiniGPT4-Video-[90f] [5] | 0.70 | 0.76 | 0.55 | 0.54 | 1.44 | 0.62 | 0.62 | 1.03 | 1.05 | 0.62 | 0.82 | 0.85 |
| MovieLLM-[1fps] [49] | 0.87 | 0.95 | 0.82 | 0.70 | 0.15 | 0.81 | 0.52 | 1.12 | 1.22 | 0.54 | 1.05 | 0.97 |
| PLLaVA-7B-[16f] [58] | 1.03 | 1.08 | 1.06 | 0.86 | 0.52 | 1.02 | 0.64 | 1.25 | 1.17 | 0.98 | 1.01 | 1.03 |
| ShareGPT4Video-8B-[16f*] [12] | 1.05 | 1.20 | 1.05 | 1.00 | 0.32 | 1.04 | 0.89 | 1.06 | 1.19 | 1.01 | 0.99 | 1.03 |
| VideoStreaming-[64f+] [46] | 1.12 | 1.38 | 1.13 | 0.8 | 0.32 | 1.13 | 0.77 | 1.27 | 1.11 | 1.01 | 1.10 | 1.09 |
| LLaVA-NeXT-Video-[32f] [64] | **1.14** | 1.35 | 1.15 | 0.97 | 0.58 | **1.14** | 0.64 | 1.38 | 1.30 | 1.27 | 1.03 | **1.13** |
| *Open-Source LVLMs for Images* | | | | | | | | | | | | |
| Idefics2-8B-[1f] [28] | 0.95 | 1.06 | 0.85 | 0.81 | 1.35 | 0.90 | 0.73 | 1.14 | 1.08 | 1.09 | 1.04 | 1.03 |
| Idefics2-8B-[8f] | 1.10 | 1.23 | 1.07 | 0.89 | 0.77 | 1.06 | 0.77 | 1.27 | 1.41 | 1.11 | 1.14 | 1.16 |
| Qwen-VL-Chat-[1f] [6] | 0.60 | 0.72 | 0.59 | 0.53 | 1.16 | 0.63 | 0.58 | 0.60 | 0.54 | 0.53 | 0.47 | 0.53 |
| Qwen-VL-Chat-[8f] | 0.52 | 0.44 | 0.62 | 0.33 | 0.15 | 0.53 | 0.45 | 0.59 | 0.50 | 0.36 | 0.37 | 0.45 |
| mPLUG-Owl2-[1f] [60] | 0.85 | 1.05 | 0.79 | 0.79 | 0.68 | 0.83 | 0.54 | 1.06 | 1.05 | 0.74 | 0.83 | 0.86 |
| mPLUG-Owl2-[8f] | 1.15 | 1.34 | 1.18 | 0.99 | 0.27 | 1.15 | 0.63 | 1.33 | 1.30 | 1.03 | 1.11 | 1.11 |
| InternVL-Chat-v1.5-[1f] [13] | 0.84 | 0.98 | 0.72 | 0.78 | 1.44 | 0.80 | 0.57 | 1.02 | 1.12 | 0.83 | 0.88 | 0.90 |
| InternVL-Chat-v1.5-[8f] | 1.26 | 1.51 | 1.22 | 1.01 | 1.21 | 1.25 | 0.88 | 1.40 | 1.48 | 1.28 | 1.09 | 1.22 |
| InternVL2-26B-[16f] | 1.41 | 1.56 | 1.48 | 1.23 | 0.52 | 1.42 | 1.06 | 1.61 | 1.45 | 1.38 | 1.23 | 1.35 |
| VILA1.5-13B-[14f] [35] | 1.36 | 1.51 | 1.45 | 1.26 | 0.24 | 1.39 | 0.80 | 1.52 | 1.30 | 1.40 | 1.28 | 1.28 |
| VILA1.5-40B-[14f] | **1.61** | 1.78 | 1.72 | 1.35 | 0.47 | **1.63** | 1.12 | 1.78 | 1.61 | 1.48 | 1.45 | **1.52** |
| *Proprietary LVLMs for Images* | | | | | | | | | | | | |
| Claude-3v-Opus-[4f] [4] | 1.19 | 1.37 | 1.11 | 1.00 | 1.56 | 1.16 | 1.12 | 1.35 | 1.36 | 1.17 | 1.05 | 1.20 |
| Gemini-Pro-v1.0-[8f] [51] | 1.49 | 1.72 | 1.50 | 1.28 | 0.79 | 1.49 | 1.02 | 1.66 | 1.58 | 1.59 | 1.40 | 1.45 |
| Gemini-Pro-v1.0-[16f] | 1.48 | 1.61 | 1.56 | 1.30 | 0.65 | 1.50 | 1.15 | 1.57 | 1.55 | 1.36 | 1.33 | 1.39 |
| Gemini-Pro-v1.5-[8f] [51] | 1.30 | 1.51 | 1.30 | 0.98 | 2.03 | 1.32 | 1.06 | 1.62 | 1.36 | 1.25 | 0.94 | 1.22 |
| Gemini-Pro-v1.5-[16f] | 1.60 | 1.81 | 1.59 | 1.60 | 2.00 | 1.61 | 1.58 | 1.77 | 1.69 | 1.80 | 1.24 | 1.55 |
| Gemini-Pro-v1.5-[1fps] | 1.94 | 1.99 | 2.04 | 1.70 | 1.90 | 1.98 | 1.98 | 2.02 | 1.92 | 1.78 | 1.63 | 1.86 |
| GPT-4v-[8f] [42] | 1.53 | 1.68 | 1.45 | 1.43 | 1.79 | 1.51 | 1.14 | 1.81 | 1.70 | 1.59 | 1.39 | 1.52 |
| GPT-4v-[16f] | 1.68 | 1.83 | 1.65 | 1.40 | 1.76 | 1.66 | 1.45 | 1.91 | 1.86 | 1.83 | 1.53 | 1.69 |
| GPT-4o-[1f] [43] | 0.70 | 0.99 | 0.61 | 0.53 | 2.19 | 0.73 | 0.47 | 0.82 | 0.63 | 0.69 | 0.44 | 0.59 |
| GPT-4o-[8f] | 1.62 | 1.82 | 1.59 | 1.43 | 1.95 | 1.63 | 1.33 | 1.89 | 1.60 | 1.60 | 1.44 | 1.57 |
| GPT-4o-[16f] | 1.86 | 2.03 | 1.88 | 1.67 | 2.13 | 1.89 | 1.78 | 1.95 | 1.78 | 1.90 | 1.68 | 1.80 |
| GPT-4o-[1fps] | **2.15** | 2.23 | 2.24 | 2.01 | 1.90 | **2.19** | 2.11 | 2.12 | 2.17 | 1.94 | 1.97 | **2.08** |

Table 3: **Evaluation Result of Video Models on MMBench-Video.** CP, FP[S], FP[C], HL stands for four L-2 perception capabilities: Coarse Perception, Single-Instance Finegrained Perception, Cross-Instance Finegrained Perception, and Hallucination. LR, AR, RR, CSR, TR stand for five reasoning capabilities: Logic, Attribute, Relation, Commonsense, and Temporal Reasoning. All scores are based on a 3-grade marking scheme: 0 for worst, 3 for best. -[$N$f] indicates the method take $N$ frames uniformly sampled from a video as input. -[$N$fps] indicates the method uses $N$ frames per second uniformly sampled from a video as input. Among *Open-Source Video-LLMs*, ShareGPT4Video-8B-[16f*] follows the IG-VLM [27] strategy and presents 16 frames in a $4 \times 4$ grid. VideoStreaming-[64f+] accepts streaming videos and takes at least 64 frames as inputs.

| Model | MMBench | | | | | | | MMStar | | | | | | |
|---|---|---|---|---|---|---|---|---|---|---|---|---|---|---|
| | FP-S | FP-C | CP | LR | AR | RR | Overall | CP | FP | IR | LR | Math | ST | Overall |
| *Open-Source Video-LLMs* | | | | | | | | | | | | | | |
| Video-ChatGPT | 41.87 | 27.37 | 32.87 | 13.71 | 53.05 | 30.46 | 34.50 | 40.80 | 24.80 | 36.00 | 26.00 | 28.00 | 22.40 | 29.67 |
| Video-LLaVA | 57.44 | 42.46 | 62.98 | 14.52 | 68.90 | 43.10 | 52.32 | 55.20 | 20.40 | 37.60 | 25.20 | 25.60 | 24.00 | 31.33 |
| Chat-UniVi | 47.75 | 35.75 | 57.18 | 9.68 | 62.19 | 33.91 | 45.04 | 50.00 | 30.80 | 42.80 | 30.40 | 30.00 | 24.40 | 34.73 |
| VideoChat2 | 42.91 | 30.72 | 54.14 | 7.26 | 54.88 | 32.18 | 41.02 | 47.60 | 22.80 | 32.80 | 27.20 | 26.40 | 13.20 | 28.33 |
| PLLaVA-7B | 59.17 | 40.78 | 60.50 | 17.74 | 58.54 | 58.05 | 52.79 | 53.60 | 34.40 | 40.80 | 32.40 | 30.00 | 17.20 | 34.73 |
| *Open-Source LVLMs for Images* | | | | | | | | | | | | | | |
| MiniCPM-V-2 | 78.89 | 50.84 | 72.93 | 26.61 | 75.00 | 65.52 | 66.02 | 58.00 | 32.40 | 50.00 | 38.40 | 32.80 | 22.80 | 39.07 |
| LLaVA-v1.5-7B | 69.90 | 56.98 | 70.17 | 25.81 | 67.07 | 53.45 | 61.38 | 57.20 | 24.40 | 41.60 | 28.40 | 26.40 | 20.40 | 33.07 |
| InternVL-Chat-v1.5 | 88.58 | 73.18 | 80.94 | 58.06 | 85.98 | 80.46 | 79.95 | 70.40 | 52.80 | 65.20 | 58.40 | 56.00 | 39.60 | 57.07 |
| Idefics2-8B | 81.31 | 65.36 | 73.20 | 41.94 | 80.49 | 76.44 | 72.29 | 66.00 | 42.40 | 61.60 | 49.60 | 40.00 | 37.20 | 49.47 |
| Phi-3-Vision | 78.89 | 61.45 | 76.80 | 47.58 | 79.27 | 74.14 | 72.29 | 60.00 | 38.80 | 59.20 | 45.20 | 42.40 | 40.80 | 47.73 |

Table 4: **Comparison of Image Models and Video Models on MMBench and MMStar.** We follow the official practice to perform evaluation on these two benchmarks. For MMBench, we report the results on MMBench-DEV-EN-v1.1. We adopt the abbreviations for capabilities that are defined in the original papers.

| Model | Subtitle | Overall Mean | Perception | | | | | Reasoning | | | | | |
|---|---|---|---|---|---|---|---|---|---|---|---|---|---|
| | | | CP | FP-S | FP-C | HL | Mean | LR | AR | RR | CSR | TR | Mean |
| GPT-4o | ✗ | 0.29 | 0.03 | 0.10 | 0.08 | 1.84 | 0.16 | 0.38 | 0.51 | 0.46 | 0.18 | 0.79 | 0.54 |
| | ✔ | 1.22 | 1.17 | 1.18 | 0.87 | 1.90 | 1.17 | 0.69 | 1.40 | 1.33 | 0.71 | 1.62 | 1.31 |
| GPT-4o-[8f] | ✗ | 1.66 | 1.90 | 1.63 | 1.52 | 1.88 | 1.68 | 1.61 | 1.87 | 1.50 | 1.52 | 1.65 | 1.63 |
| | ✔ | 1.94 | 1.98 | 1.92 | 1.71 | 1.72 | 1.90 | 2.05 | 2.07 | 1.97 | 2.00 | 1.99 | 2.01 |
| GPT-4o-[16f] | ✗ | 1.90 | 2.13 | 1.86 | 1.56 | 2.20 | 1.90 | 2.26 | 2.04 | 1.47 | 2.00 | 1.85 | 1.91 |
| | ✔ | 2.05 | 2.14 | 2.07 | 1.79 | 1.64 | 2.04 | 2.31 | 2.22 | 1.78 | 2.33 | 2.05 | 2.12 |

Table 5: **GPT-4o's performance can be further improved by incorporating YouTube generated subtitles.** We report the performance on a subset of MMBench-Video, for which the auto generated subtitles are available.

**Proprietary LVLMs for Images.** Unlike their open-source counterparts, most proprietary LVLMs accept arbitrary interleaved images and text as input. We evaluate several proprietary LVLMs, including Claude-3v, Gemini-Pro-v[1.0/1.5], GPT-4v, and GPT-4o on MMBench-Video with varying numbers of frames. Claude-3v struggles with 8-frame inputs and is only evaluated under the 4-frame setting. As anticipated, it exhibits the poorest performance among proprietary LVLMs when handling multiple frames. In contrast, other proprietary models demonstrate notably superior performance compared to the state-of-the-art open-source VILA-1.5. Particularly impressive is GPT-4o, which, when processing 16 frames, achieves an outstanding overall score of **1.86**. This result positioned GPT-4o 63% ahead of the best open-source video LLM and 16% ahead of the best open-source image LVLM. To assess the potential of advanced proprietary models, we experiment with the fps sampling method. By increasing the number of video frames, the model's perception improves as adjacent frames provide mutual support, resulting in more accurate interpretations. This approach also captures previously missed content, enhancing the model's ability to answer questions reliant on such information and boosting its reasoning skills. The only observed drawback is a slight increase in hallucination, possibly due to excessive video content leading to responses not grounded in reality.

## 4.2 Performance of Video-LLMs on Image VQA Benchmarks

Intuitively, a Video-LLM is expected to not only possess all capabilities of an image-based LVLM but also exhibit video-specific competencies, such as future prediction or causal reasoning. In light of the underwhelming performance of Video-LLMs on MMBench-Video, we broaden our evaluation to include image VQA benchmarks to determine if these models have the necessary skills for comprehending static content. We evaluate five Video-LLMs on two extensive image VQA benchmarks: MMBench [38] and MMStar [13]. To accommodate the input format of Video-LLMs, we create pseudo video clips by duplicating static frames, which then serves as the input for the evaluation. In Tab. 4, we list the performance of Video-LLMs alongside several representative image LVLMs for comparative analysis. On both benchmarks, existing Video-LLMs exhibit subpar performance. Notably, top-performing Video-LLMs such as PLLaVA and Video-LLaVA show performance that is either on par with or inferior to LLaVA-v1.5-7B, a rudimentary baseline for image multimodal understanding, and significantly trail behind the state-of-the-art image LVLM,

| Judge Model | LVLM | Video-LLaVA | GPT-4o |
|---|---|---|---|
| GPT-3.5-Turbo | 1106 | 2.09 | 2.45 |
| | 0613 | 1.80 | 2.11 |
| GPT-4-Turbo | 1106 | 1.05 | 1.62 |
| | 0125 | 0.90 | 1.61 |
| Qwen2-72B-Instruct | | 1.15 | 1.80 |

Table 6: **Evaluation results obtained with different GPT judges on MMBench-Video.** The overall mean scores are reported.

| Judge Model | LVLM | Video-LLaVA | GPT-4o |
|---|---|---|---|
| GPT-3.5-Turbo | 1106 | 0.98 | 0.815 |
| | 0613 | 0.89 | 0.685 |
| GPT-4-Turbo | 1106 | 0.36 | 0.295 |
| | 0125 | 0.36 | 0.255 |
| Qwen2-72B-Instruct | | 0.41 | 0.320 |

Table 7: **The mean absolute error (MAE) of different GPT Judges with human preferences on a randomly selected subset.**

InternVL-v1.5. This evaluation underscores the current limitations in the spatial understanding capabilities of Video-LLMs.

### 4.3 Incorporating Speech Further Improves Proprietary LVLMs

Video inherently comprises both visual and audio signals. However, the majority of existing LVLMs for video understanding predominantly focus on visual features, often neglecting the valuable information embedded in audio signals. To explore the potential impact of integrating audio features on video understanding, we conducted experiments using video title tracks (VTT) sourced from YouTube, which are automatically generated through speech recognition techniques. We incorporate these subtitles into the prompt as supplementary context. Experimental results in Tab. 5 reveal that the inclusion of audio/speech information enhances the performance of the state-of-the-art proprietary model, GPT-4o. The subtitles offer a rich source of high-density information, facilitating the LLM's ability to accurately address the questions, thereby leading to a comprehensive performance improvement. Nonetheless, the increased information richness also heightens the risk of hallucinations, where the model may produce responses about non-existent content. The effectiveness hinges on the information density and the level of redundancy, necessitating a careful balance in applications.

### 4.4 The Superior Performance of GPT-4 as a Judge

Due to the discrepancy between the predictions of Video-LLM and the ground truth answers, existing VideoQA benchmarks largely rely on a judge model to assess the model responses. The capability of judge models can significantly influence the final results. To quantitatively study the impact of judge models, we utilize different versions of GPT-3.5 and GPT-4 for evaluation and report the results in Tab. 6. We observe that GPT-3.5 tends to assign higher scores (typically 2 and 3), which can result in inflated final results and potential inaccuracies. To investigate the alignment with human preferences across different judge models, we conduct study based on a randomly selected subset of 100 questions. Two of the authors manually rate the responses from Video-LLaVA and GPT-4o, and we then report the mean absolute error between different judge models and the averaged human ratings. Tab. 7 shows that GPT-3.5 exhibits a significantly larger discrepancy with human preferences and greater inter-version variance. In contrast, Qwen2-72B-Instruct aligned more closely with human ratings, suggesting the potential of using advanced open-source LLMs as evaluators. Compared to the other two, GPT-4 demonstrated greater resistance to manipulation and fairly evaluated predictions without bias, supporting its role as an evaluator for the MMBench-Video benchmark.

## 5 Conclusion

This work introduces MMBench-Video, a novel long-form, multi-shot VideoQA benchmark specifically designed to evaluate the capabilities of LVLMs in understanding video content. MMBench-Video encompasses a diverse range of video topics and fine-grained capabilities. Extensive evaluations on MMBench-Video allow us to identify significant performance limitations among existing Video-LLMs in both spatial and temporal understanding.

## Acknowledgement

This project is supported by the National Key R&D Program of China (No.2022ZD0161600), the Shanghai Postdoctoral Excellence Program (No.2023023), China Postdoctoral Science Fund (No.2024M751559), and Shanghai Artificial intelligence Laboratory. This project is funded in part by the Centre for Perceptual and Interactive Intelligence (CPII) Ltd under the Innovation and Technology Commission (ITC)'s InnoHK. Dahua Lin is a PI of CPII under the InnoHK.

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

# A  OpenSource Datasets and Codes

We have uploaded the full MMBench-Video dataset to HuggingFace, you can access the data at `https://huggingface.co/datasets/opencompass/MMBench-Video`. The code related to performance evaluation of Video-LLM and LVLM using MMBench-Video has been released in VLMEvalKit and the test results have been published in OpenVLM Video Leaderboard. We provide the DataSheet at the end of this document.

**Author Statement and Data Licence.**    The authors bear all responsibility in case of violation of rights and confirm that this dataset is opensourced under the Attribution-NonCommercial 4.0 International (CC BY-NC 4.0) license.

# B  Additional Experiments

## B.1  The Impact of Incorporating Speech over the entire MMBench-Video

In the main paper, we report the quantitative results of speech improvement on the subset of videos with subtitles available from YouTube. In MMBench-Video, approximately half of videos do not include parseable video title tracks. In Tab. 8, we report the impact of incorporating speech information across the entire MMBench-Video dataset. Initially, we tested GPT-4o without visual input, and the results showed that even advanced models struggle to solve most problems using only question inputs. However, with subtitles, the model's ability to visualize content improves, boosting performance by 200%. Further experiments with 8 and 16 frame inputs show that, even though only half of the VideoQAs include speech, the overall performance on the full benchmark remains significantly enhanced. While only half of the VideoQAs are enhanced with speech, the overall performance improvement on the full benchmark remains significant. It is evident that the enhancement in reasoning capabilities surpasses that of perceptual abilities. Speech typically conveys contextual information absent in static visual inputs, facilitating further reasoning by the VLM. Meanwhile, improvements in coarse perception are minimal or remain largely unchanged (for GPT-4o-[16f]). This can be attributable to the fact that perception is predominantly reliant on visual inputs, and speech information does not significantly augment the model's performance in coarse perception.

| Model | Subtitle | Overall Mean | Perception | | | | | Reasoning | | | | | |
|---|---|---|---|---|---|---|---|---|---|---|---|---|---|
| | | | CP | FP-S | FP-C | HL | Mean | LR | AR | RR | CSR | TR | Mean |
| GPT-4o | ✘ | 0.25 | 0.03 | 0.11 | 0.07 | 1.82 | 0.16 | 0.39 | 0.55 | 0.32 | 0.30 | 0.55 | 0.45 |
| | ✔ | 0.74 | 0.60 | 0.64 | 0.50 | 2.11 | 0.67 | 0.46 | 0.98 | 0.91 | 0.44 | 0.94 | 0.83 |
| GPT-4o-[8f] | ✘ | 1.62 | 1.82 | 1.59 | 1.43 | 1.95 | 1.63 | 1.33 | 1.89 | 1.60 | 1.60 | 1.44 | 1.57 |
| | ✔ | 1.79 | 1.90 | 1.82 | 1.51 | 1.82 | 1.79 | 1.57 | 1.98 | 1.81 | 1.81 | 1.71 | 1.78 |
| GPT-4o-[16f] | ✘ | 1.86 | 2.03 | 1.88 | 1.67 | 2.13 | 1.89 | 1.78 | 1.95 | 1.78 | 1.90 | 1.68 | 1.80 |
| | ✔ | 1.96 | 2.03 | 2.00 | 1.77 | 1.89 | 1.97 | 1.87 | 1.98 | 1.92 | 1.99 | 1.84 | 1.90 |

Table 8: **GPT-4o's performance can be further improved by incorporating YouTube generated subtitles even under whole dataset.**

## B.2  Detailed Analysis of L-2 Capability

Based on Tab.3, it is evident that hallucination is the most significant limitation in L-2 perceptual capabilities for all Video-LLMs, in contrast to the state-of-the-art proprietary LVLMs. This indicates that existing Video-LLMs are unable to dismiss questions pertaining to videos when uncertain and are inclined to generate answers for questions regarding non-existent visual content.

Regarding LVLMs, the number of frames significantly influences the performance of most L-2 capabilities in MMBench-Video. With an increase in the number of frames, the enhancement in perceptual capabilities becomes more pronounced than that in reasoning capabilities. Owing to a more extensive training corpus and superior safety mechanisms, proprietary LVLMs exhibit superior performance in challenging capabilities such as logical reasoning, commonsense reasoning, and hallucination.

Interestingly, despite Idefics2-8B-[1f] utilizing a single image as input, it still outperforms all Video-LLMs in temporal reasoning tasks. This suggests that Video-LLMs are not effectively leveraging

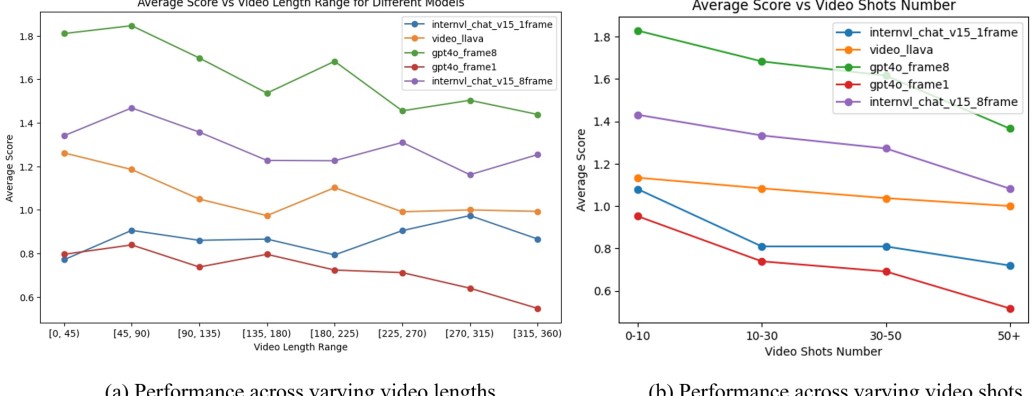

(a) Performance across varying video lengths      (b) Performance across varying video shots

Figure 6: **Performance across varying video lengths and video shots**. We report the results of InternVL-Chat-V1.5-[1/8f], GPT-4o-[1/8f] and Video-LLaVA judged by GPT-4-Turbo (1106).

diverse temporal information, underscoring the necessity to enhance the diversity of instruction tuning data for these models.

## B.3 Model Performance across Video Time Length and Video Shots

The length of the video and the number of shots are indeed key factors affecting model performance. Videos with fewer and shorter shots may perform better at lower frame counts, while longer or multi-shot videos require more visual content to fully leverage the model's capabilities.

Fig. 6 illustrates the trend of model scores in relation to shot count and video length. It is evident that the performance of GPT-4o, sampled at different frame rates, declines as video length increases, whereas the performance of open-source models such as InternVL-Chat-v1.5 and Video-LLaVA remains relatively stable. Compared to video length, model performance is more significantly influenced by the number of video shots. With over 50 shots in a video, the performance of GPT-4o drops to 75% of its original score. This suggests that the model's performance is more closely tied to the number of shots than to video length, as the frequent shot transitions make it more challenging for the model to comprehend the video, resulting in lower scores.

## C Additional Dataset Analysis

In this section, we present more details about the MMBench-Video dataset: including the technique we adopted to filter temporal dispensable questions and some statistics on the linguistic characteristics of questions in MMBench-Video. In Figs. 8 to 10, we display a selection of samples from MMBench-Video, showcasing videos, images, questions, and reference answers for illustrative purposes.

### C.1 Temporal Dispensable Data Filtering with LVLM

To ensure that the majority of questions in MMBench-Video are temporally indispensable, we implement an LVLM-based filtering process and subsequently conduct manual verification. Specifically, we employ GPT-4v, one of the most potent LVLMs, to filter out questions exhibiting high temporal irrelevance. Utilizing four distinct random seeds, we sample a single random frame as visual input for each individual VideoQA instance and conduct the inference four times. Subsequently, we utilize GPT-4 to evaluate the responses and compute the average score for each question. Questions with an average score of 2.5 or higher across the four responses were excluded from the benchmark. Based on this approach, we removed a total of 246 temporally dispensable questions from the dataset.

### C.2 Question Type Analysis

Given that the majority of existing Video-QA benchmarks are characterized by a limited range of question types, which fail to adequately represent the diverse spectrum of human conversations, the question set within MMBench-Video has been meticulously curated to encompass a wide variety of

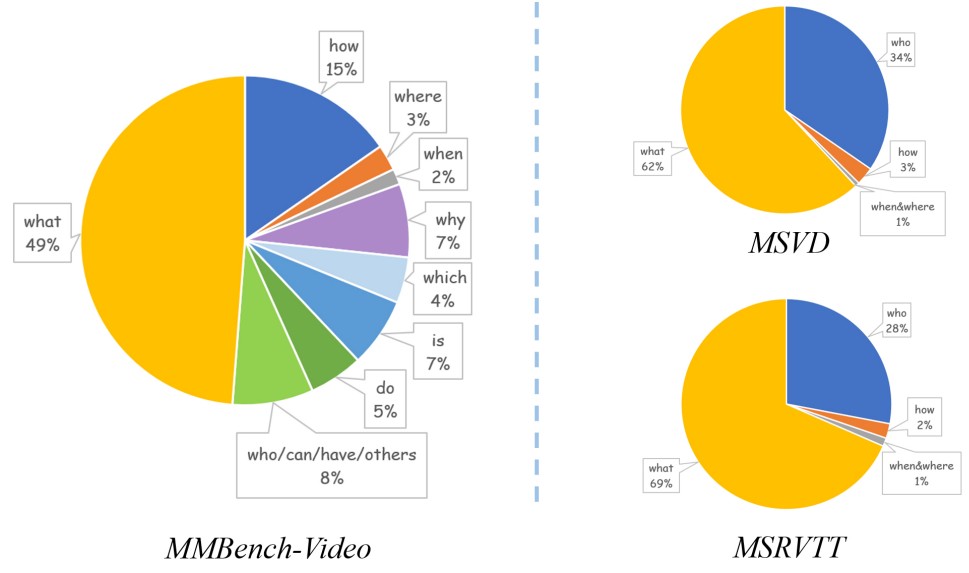

Figure 7: **Comparison of question type distribution with MSVD and MSRVTT.** MMBench-Video encompasses a more extensive assortment of question types and exhibits a distribution that is more equitably balanced among these various categories.

categories. We visualize the comparison of the question type distribution between MMBench-Video and other popular benchmarks in Fig. 7. In addition to the conventional question archetypes, namely *'what', 'who', 'how', 'when', and 'where'*, MMBench-Video extends its corpus to include additional interrogatives such as *'why', 'which', 'is / are', and 'does / do'*. The expansion diversifies the dataset and closely aligns with the style of natural human dialogues. Meanwhile, the question type distribution within MSVD or MSRVTT exhibits a significant skew. The category of 'what' predominates, comprising over 60% of the questions, in stark contrast to the significantly underrepresented categories such as 'when' and 'where', totaling a mere 1%. Nevertheless, the question type distribution within MMBench-Video has been deliberately engineered to achieve a greater equilibrium. While the 'what' category maintains its status as the most prevalent, the remaining question types are evenly distributed across various interrogative forms.

## D  Prompts Adopted in MMBench-Video

In Sec. 3.1, we outline the LLM-involved evaluation paradigm we employ, utilizing GPT-4 for scoring. The evaluation is conducted through prompts configured with a 3-grade marking (0, 1, 2, 3). In this section, we elaborate on the specific prompts utilized in the evaluation process.

**System Prompt for GPT-based Evaluation.**

```
As an AI assistant, your task is to evaluate a candidate answer in comparison to a
given correct answer.  The question itself, the correct 'groundtruth' answer, and
the candidate answer will be provided to you.  Your assessment should range from
0 to 3, based solely on the semantic similarity between the groundtruth and the
candidate answer, disregarding any grammatical differences.  A rating of 0 suggests
no similarity, implying the candidate answer is entirely incorrect.  A rating of
1 suggests low similarity, meaning the candidate answer is largely incorrect.  A
rating of 2 suggests high similarity, meaning the candidate answer is largely
correct.  Lastly, a rating of 3 indicates complete similarity, which means the
candidate answer is entirely correct.  Your response should be a single integer from
0, 1, 2, or 3.
Question:  [QUESTION]
Groundtruth answer:  [ANNOTATED ANSWER]
Candidate answer:  [CANDIDATE ANSWER]
Your response:
```

**System Prompt for the Inference of LVLMs with Multi-Frame Inputs.**

```
You will be provided with [FRAME NUM] separate frames uniformly sampled from a
video, the frames are provided in chronological order of the video.
Please analyze these images and provide the answer / answers to the following
question / questions about the video content.
If multiple questions are provided (with indices I1, I2, I3, ...), you should
organize your answers in the following json format:
{
    'I1':  'Answer to Question I1',
    'I2':  'Answer to Question I2',
    ...
}
Otherwise, please directly reply with your response to the only question.
Even if the information in these separate frames is not enough to give an answer,
PLEASE TRY YOUR BEST TO GUESS A CLEAR OR VAGUE ANSWER WHICH YOU THINK WOULD BE THE
MOST POSSIBLE ONE BASED ON THE QUESTION.
Minimize negative responses such as 'not possible to determine'.  STIMULATE YOUR
POTENTIAL AND IMAGINATION!
```

**Input Prompt Template for the Inference of LVLMs with Multi-Frame Inputs.**

```
[System Prompt]
[Subtitle (Optional):  {
    't0' - 't1':  subtitle 1,
    't1' - 't2':  subtitle 2,
    ......
}]
[Multi-Frame Inputs]
[Question Set:  {
    'index 1':  question 1 for this video,
    'index 2':  question 2 for this video,
    ......
}]
```

# E   Limitations and Broader Impacts

**Limitations.**  In this study, we introduce MMBench-Video, a novel long-form multi-shot VideoQA benchmark, and perform a comprehensive evaluation based on this benchmark. In light of budget constraints, our evaluation is focused on a curated selection of representative open-source and proprietary VLMs, which may not encompass all those most recent high-performing models. GPT-4 is adopted as a more advanced judge model for scoring the responses, while further experiments should be conducted in the future to study the feasibility of using state-of-the-art open-source LLMs as the judge. Taking into account the limited capabilities of existing Video-LLMs, we currently set the upper duration limit of videos to 6 minutes, refraining from scaling to tens of minutes or hours. The evaluation results indicate that, even with relatively modest video durations, MMBench-Video presents a significant challenge to existing Video-LLMs.

**Broader Impacts.**  As an evaluation benchmark, MMBench-Video offers detailed insights into the fine-grained capabilities of diverse vision-language models (VLMs) in the domain of video understanding, providing valuable insights for future model optimization. The new benchmark exhibits enhanced quality and enriched diversity, and employs a more precise scoring strategy, which collectively contribute to comprehensive and reliable evaluation outcomes. Leveraging MMBench-Video and other Image VQA benchmarks, we conduct a comprehensive evaluation of existing video-LLMs, revealing their limited capabilities in both spatial and temporal understanding. Additionally, MMBench-Video, being a small-scale benchmark, may not encompass every video topic and fine-grained capability. There is a risk that MMBench-Video may not adequately reflect the video understanding capabilities of VLMs in specific tasks or scenarios. We encourage users to carefully consider the intended use cases of VLMs when utilizing MMBench-Video for evaluation.

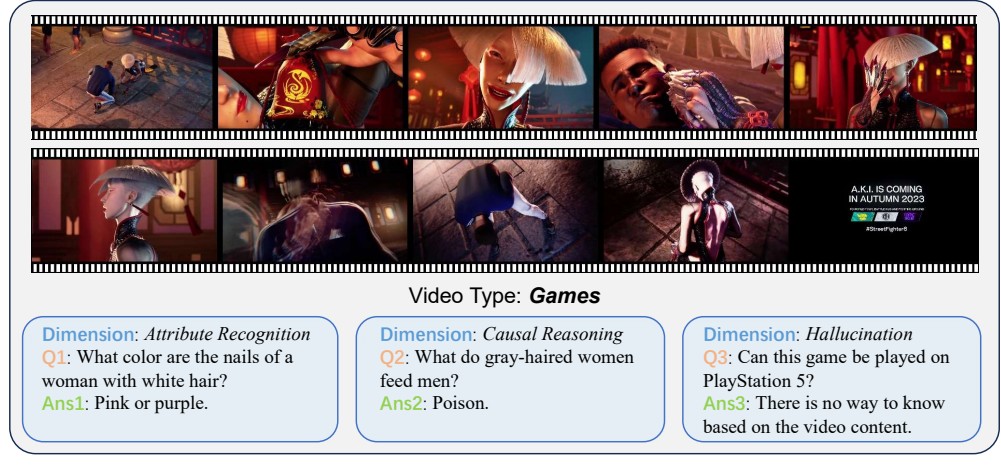

Video Type: **Games**

Dimension: *Attribute Recognition*
Q1: What color are the nails of a woman with white hair?
Ans1: Pink or purple.

Dimension: *Causal Reasoning*
Q2: What do gray-haired women feed men?
Ans2: Poison.

Dimension: *Hallucination*
Q3: Can this game be played on PlayStation 5?
Ans3: There is no way to know based on the video content.

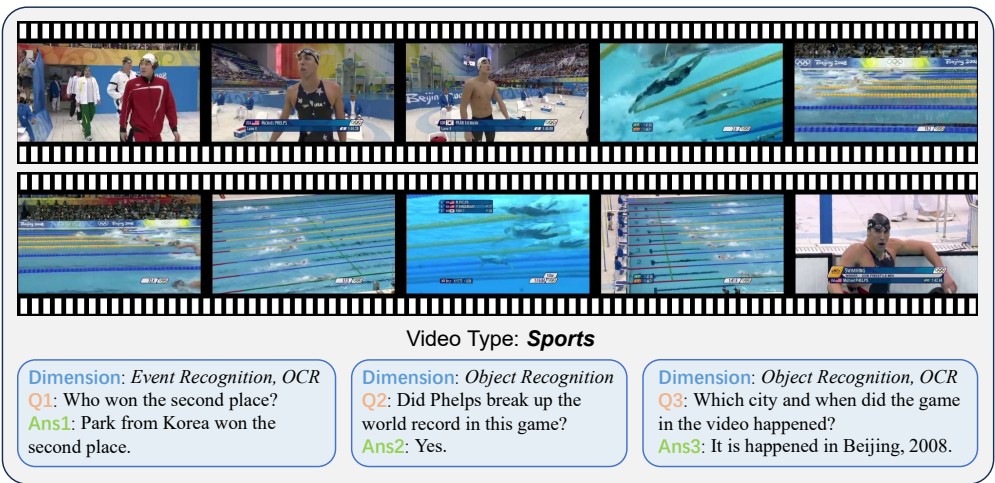

Video Type: **Sports**

Dimension: *Event Recognition, OCR*
Q1: Who won the second place?
Ans1: Park from Korea won the second place.

Dimension: *Object Recognition*
Q2: Did Phelps break up the world record in this game?
Ans2: Yes.

Dimension: *Object Recognition, OCR*
Q3: Which city and when did the game in the video happened?
Ans3: It is happened in Beijing, 2008.

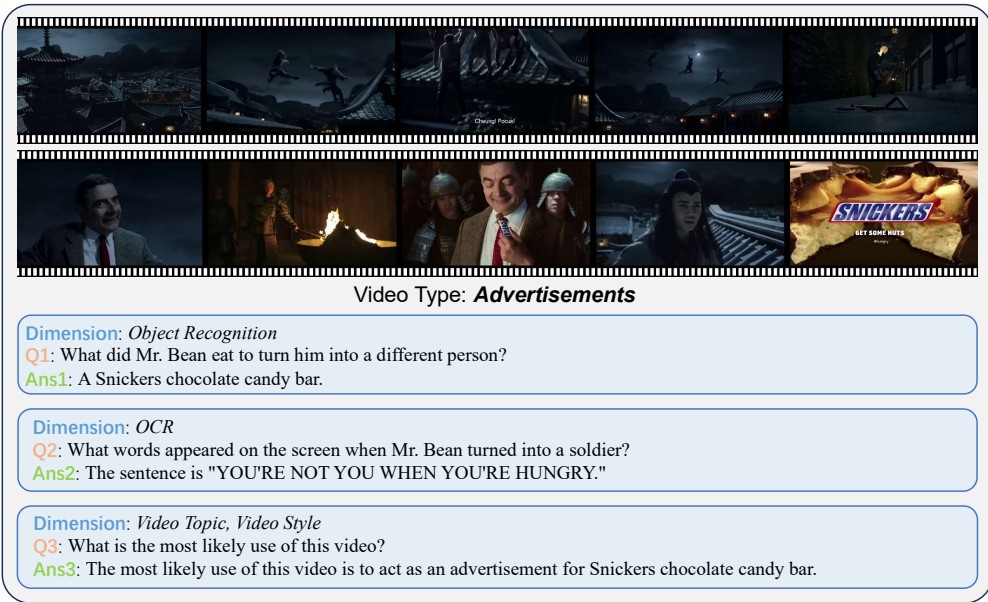

Video Type: **Advertisements**

Dimension: *Object Recognition*
Q1: What did Mr. Bean eat to turn him into a different person?
Ans1: A Snickers chocolate candy bar.

Dimension: *OCR*
Q2: What words appeared on the screen when Mr. Bean turned into a soldier?
Ans2: The sentence is "YOU'RE NOT YOU WHEN YOU'RE HUNGRY."

Dimension: *Video Topic, Video Style*
Q3: What is the most likely use of this video?
Ans3: The most likely use of this video is to act as an advertisement for Snickers chocolate candy bar.

Figure 8: **Visualization of Samples in MMBench-Video.** Part 1 out of 3.

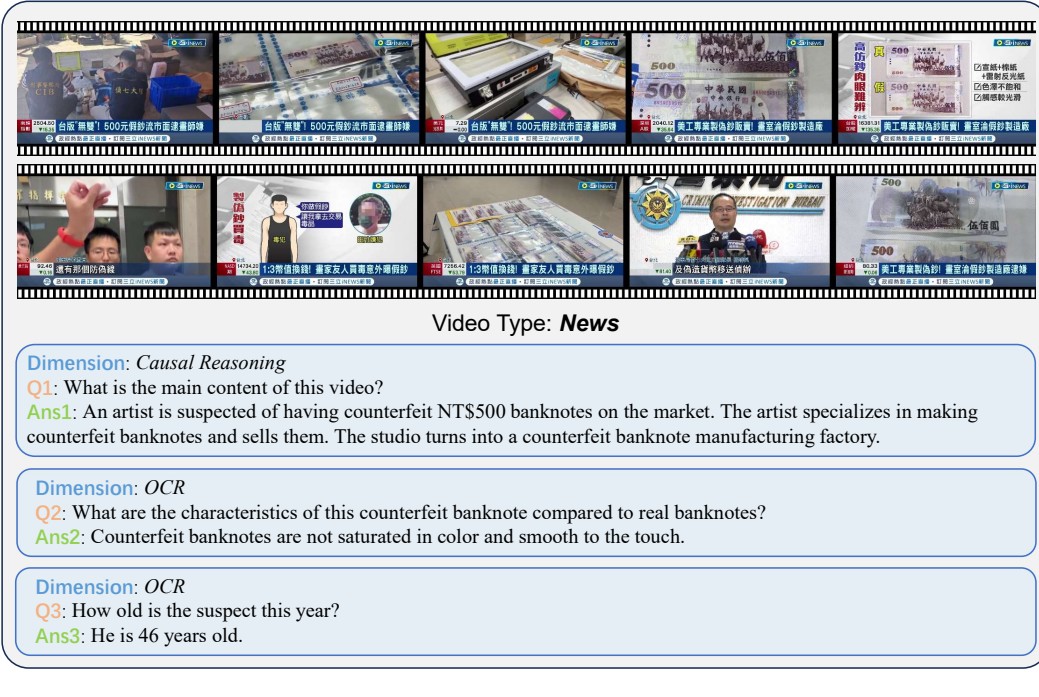

Video Type: **News**

Dimension: *Causal Reasoning*
Q1: What is the main content of this video?
Ans1: An artist is suspected of having counterfeit NT$500 banknotes on the market. The artist specializes in making counterfeit banknotes and sells them. The studio turns into a counterfeit banknote manufacturing factory.

Dimension: *OCR*
Q2: What are the characteristics of this counterfeit banknote compared to real banknotes?
Ans2: Counterfeit banknotes are not saturated in color and smooth to the touch.

Dimension: *OCR*
Q3: How old is the suspect this year?
Ans3: He is 46 years old.

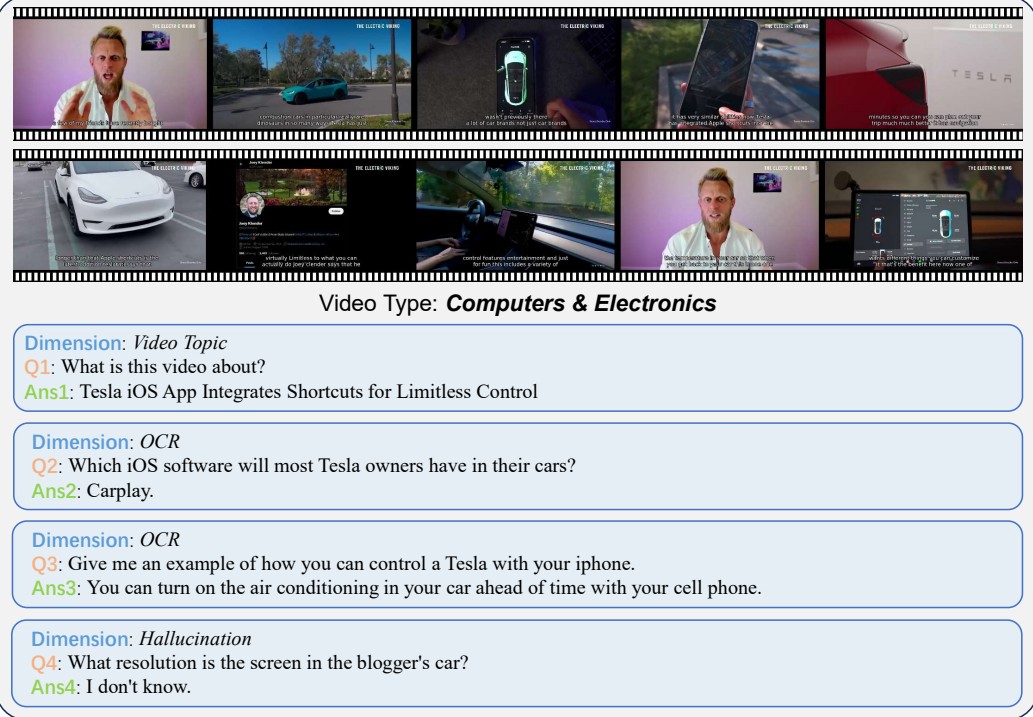

Video Type: **Computers & Electronics**

Dimension: *Video Topic*
Q1: What is this video about?
Ans1: Tesla iOS App Integrates Shortcuts for Limitless Control

Dimension: *OCR*
Q2: Which iOS software will most Tesla owners have in their cars?
Ans2: Carplay.

Dimension: *OCR*
Q3: Give me an example of how you can control a Tesla with your iphone.
Ans3: You can turn on the air conditioning in your car ahead of time with your cell phone.

Dimension: *Hallucination*
Q4: What resolution is the screen in the blogger's car?
Ans4: I don't know.

Figure 9: **Visualization of Samples in MMBench-Video.** Part 2 out of 3.

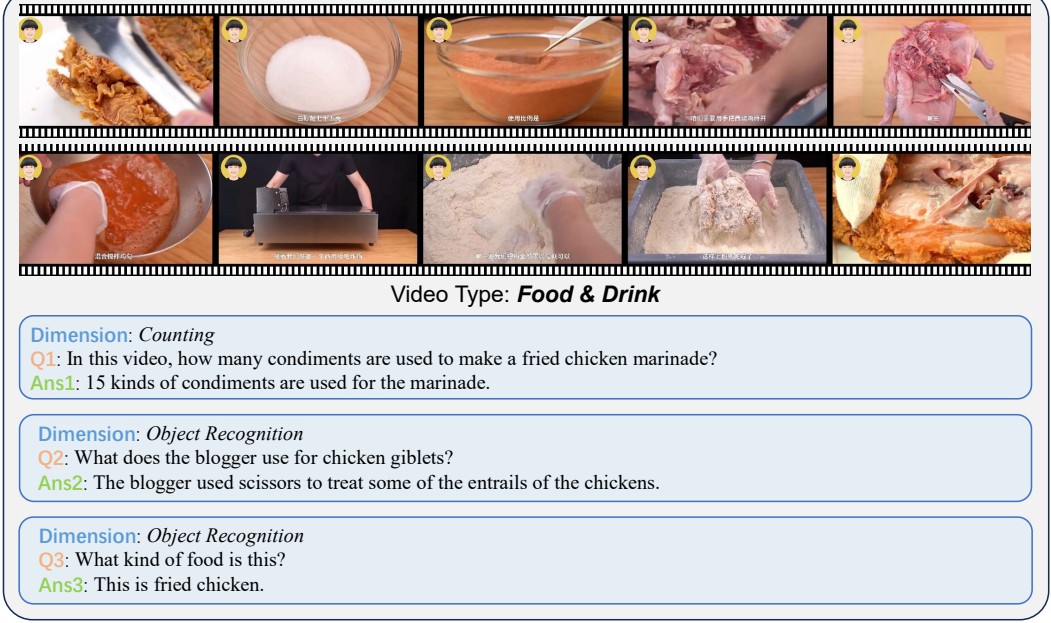

Video Type: **Food & Drink**

Dimension: *Counting*
Q1: In this video, how many condiments are used to make a fried chicken marinade?
Ans1: 15 kinds of condiments are used for the marinade.

Dimension: *Object Recognition*
Q2: What does the blogger use for chicken giblets?
Ans2: The blogger used scissors to treat some of the entrails of the chickens.

Dimension: *Object Recognition*
Q3: What kind of food is this?
Ans3: This is fried chicken.

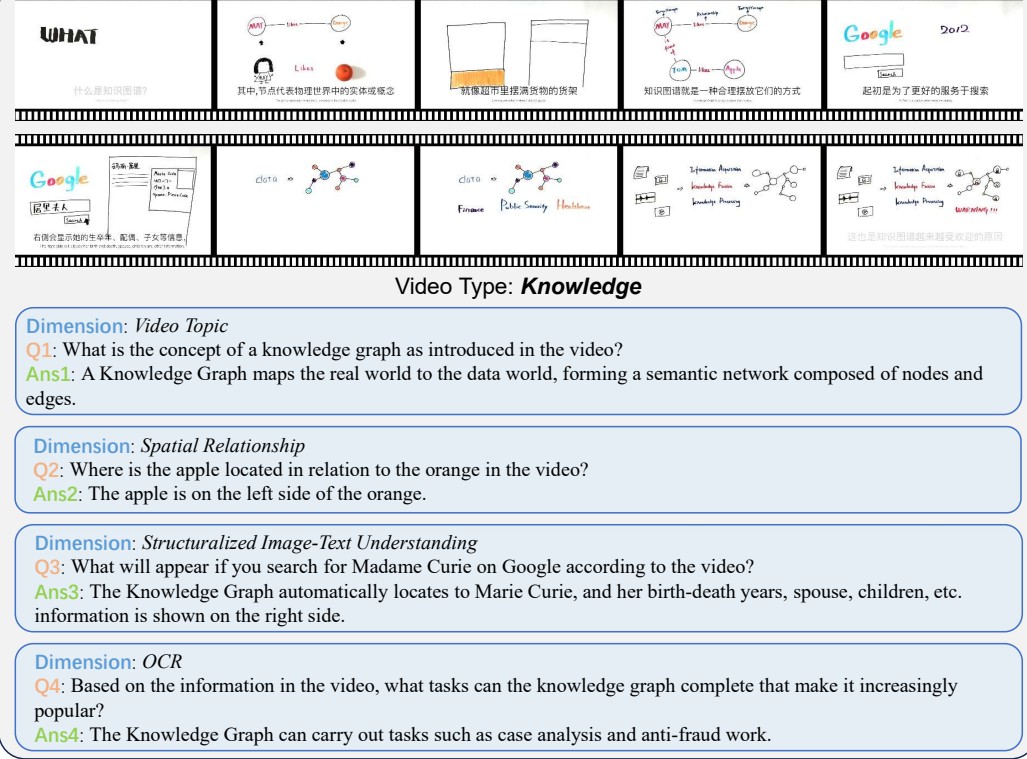

Video Type: **Knowledge**

Dimension: *Video Topic*
Q1: What is the concept of a knowledge graph as introduced in the video?
Ans1: A Knowledge Graph maps the real world to the data world, forming a semantic network composed of nodes and edges.

Dimension: *Spatial Relationship*
Q2: Where is the apple located in relation to the orange in the video?
Ans2: The apple is on the left side of the orange.

Dimension: *Structuralized Image-Text Understanding*
Q3: What will appear if you search for Madame Curie on Google according to the video?
Ans3: The Knowledge Graph automatically locates to Marie Curie, and her birth-death years, spouse, children, etc. information is shown on the right side.

Dimension: *OCR*
Q4: Based on the information in the video, what tasks can the knowledge graph complete that make it increasingly popular?
Ans4: The Knowledge Graph can carry out tasks such as case analysis and anti-fraud work.

Figure 10: **Visualization of Samples in MMBench-Video.** Part 3 out of 3.

# F  Datasheet for Datasets.

The following section contains answers to questions listed in datasheets for datas.

## 1. Motivation

a) **For what purpose was the dataset created?**
The MMBench-Video is created to evaluate the capabilities of LVLMs in understanding long-form, multi-shot video content.

b) **Who created the dataset (e.g., which team, research group) and on behalf of which entity (e.g., company, institution, organization)?**
The authors of this paper.

c) **Who funded the creation of the dataset?**
The creation of this dataset was funded by Shanghai AI Laboratory.

d) **Any other Comments?**
None.

## 2. Composition

a) **What do the instances that comprise the dataset represent (e.g., documents, photos, people, countries)?**
The MMBench-Video consists of a number of pairs of videos and corresponding questions and answers with the fine-grained video understanding capabilities they target.

b) **How many instances are there in total (of each type, if appropriate)?**
The MMBench-Video contains 1998 question-answer pairs and contains 609 videos in total.

c) **Does the dataset contain all possible instances or is it a sample (not necessarily random) of instances from a larger set?**
This is a brand-new dataset and collected from website, with manual annotation. The dataset is not samples of instances from other existing datasets.

d) **What data does each instance consist of?**
Each instance contains one video with a duration of 16.9s - 6min, a question about the video content and the corresponding answer, the category of the video, and the fine-grained video understanding capability examined by the question. Each instance can optionally contain the auto-generated subtitles sourced from YouTube, if applicable.

e) **Is there a label or target associated with each instance?**
Yes. We provide the ground-truth answer for each question.

f) **Is any information missing from individual instances?**
N/A.

g) **Are relationships between individual instances made explicit (e.g., users' movie ratings, social network links)?**
N/A.

h) **Are there recommended data splits (e.g., training, development/validation, testing)?**
N/A. The dataset is only designed for evaluation.

i) **Are there any errors, sources of noise, or redundancies in the dataset?**
N/A.

j) **Is the dataset self-contained, or does it link to or otherwise rely on external resources (e.g., websites, tweets, other datasets)?**
The dataset is self-contained.

k) **Does the dataset contain data that might be considered confidential? (e.g., data that is protected by legal privilege or by doctor-patient confidentiality, data that includes the content of individuals' non-public communications)?**
N/A.

l) **Does the dataset contain data that, if viewed directly, might be offensive, insulting, threatening, or might otherwise cause anxiety?**

N/A.

m) **Does the dataset relate to people?**

Yes. The questions and answers are annotated by human.

n) **Does the dataset identify any subpopulations (e.g., by age, gender)?**

No.

o) **Is it possible to identify individuals (i.e., one or more natural persons), either directly or indirectly (i.e., in combination with other data) from the dataset?**

No.

p) **Does the dataset contain data that might be considered sensitive in any way (e.g., data that reveals racial or ethnic origins, sexual orientations, religious beliefs, political opinions or union memberships, or locations; financial or health data; biometric or genetic data; forms of government identification, such as social security numbers; criminal history)?**

No.

q) **Any other comments?**

None.

## 3. Collection Process

a) **How was the data associated with each instance acquired?**

See main paper for details.

b) **What mechanisms or procedures were used to collect the data (e.g., hardware apparatus or sensor, manual human curation, software program, software API)?**

We collect the video data from YouTube, and get subtitles from the same website. Humans are required to propose a question and corresponding answer based on the video.

c) **If the dataset is a sample from a larger set, what was the sampling strategy (e.g. deterministic, probabilistic with specific sampling probabilities)?**

No.

d) **Who was involved in the data collection process (e.g., students, crowdworkers, contractors) and how were they compensated (e.g., how much were crowdworkers paid)?**

The authors and some undergraduate volunteers are involved in the data collection process and are paid a fair wage.

e) **Over what timeframe was the data collected?**

The dataset is collected in 2023.

f) **Were any ethical review processes conducted (e.g., by an institutional review board)?**

All videos in our benchmark are human-selected based on appropriate value propositions and undergo a second manual quality check to ensure there are no ethical violations.

g) **Does the dataset relate to people?**

Yes.

h) **Did you collect the data from the individuals in question directly, or obtain it via third parties or other sources?**

We obtained video data from the Youtube.

i) **Were the individuals in question notified about the data collection?**

We didn't collect the data from the individuals. The data was collected from public web sources instead.

j) **Did the individuals in question consent to the collection and use of their data?**

We didn't collect the data from the individuals.

k) **If consent was obtained, were the consenting individuals provided with a mechanism to revoke their consent in the future or for certain uses?**

N/A.

l) **Has an analysis of the potential impact of the dataset and its use on data subjects been conducted?**

Yes. See supplementary materials for details.

m) **Any other comments?**

None.

## 4. Preprocessing, Cleaning and Labeling

a) **Was any preprocessing/cleaning/labeling of the data done (e.g., discretization or bucketing, tokenization, part-of-speech tagging, SIFT feature extraction, removal of instances, processing of missing values)?**

Some raw YouTube videos are trimmed based on the annotations.

b) **Was the "raw" data saved in addition to the preprocessed/cleaned/labeled data?**

The "raw" data (such as untrimmed videos) are saved, and dataset users can retrieve them via YouTube IDs.

c) **Is the software used to preprocess/clean/label the instances available?**

Not applicable.

d) **Any other comments?**

None.

## 5. Uses

a) **Has the dataset been used for any tasks already?**

No.

b) **Is there a repository that links to any or all papers or systems that use the dataset?**

Not applicable.

c) **What (other) tasks could the dataset be used for?**

It also can be used to evaluate the video understanding capability of VLMs.

d) **Is there anything about the composition of the dataset or the way it was collected and preprocessed/cleaned/labeled that might impact future uses?**

None.

e) **Are there tasks for which the dataset should not be used?**

No.

f) **Any other comments?**

None.

## 6. Distribution

a) **Will the dataset be distributed to third parties outside of the entity on behalf of which the dataset was created?**

Yes, the dataset will be made publicly available.

b) **How will the dataset be distributed? (e.g., tarball on website, API, GitHub)?**

It will be distributed as a HuggingFace Dataset: `https://huggingface.co/datasets/opencompass/MMBench-Video`

c) **When will the dataset be distributed?**

It will be released in June 2024.

d) **Will the dataset be distributed under a copyright or other intellectual property (IP) license, and/or under applicable terms of use (ToU)?**
We release our benchmark under CC BY-NC 4.0 license.

e) **Have any third parties imposed IP-based or other restrictions on the data associated with the instances?**
No.

f) **Do any export controls or other regulatory restrictions apply to the dataset or to individual instances?**
No.

g) **Any other comments?**
None.

## 7. Maintenance

a) **Who will be supporting/hosting/maintaining the dataset?**
Xinyu Fang

b) **How can the owner/curator/manager of the dataset be contacted (e.g., email address)?**
fangxinyu@pjlab.org.cn

c) **Is there an erratum?**
Currently, we do not have an erratum. We will update if we find errors.

d) **Will the dataset be updated (e.g., to correct labeling errors, add new instances, delete instances)?**
Yes.

e) **If the dataset relates to people, are there applicable limits on the retention of the data associated with the instances (e.g., were individuals in question told that their data would be retained for a fixed period of time and then deleted)?**
Not applicable.

f) **Will older versions of the dataset continue to be supported/hosted/maintained?**
Yes, older versions of the benchmark will be maintained.

g) **If others want to extend/augment/build on/contribute to the dataset, is there a mechanism for them to do so?**
Yes, they can contact the maintainer via email or create a PR / issue on the github.

h) **Any other comments?**
None.

