# OpenReview forum: "MMBench-Video: A Long-Form Multi-Shot Benchmark for Holistic Video Understanding"
_NeurIPS.cc/2024/Datasets_and_Benchmarks_Track — NeurIPS 2024 Track Datasets and Benchmarks Poster_

### Official Review · Reviewer_8fpf · 2024-07-22
**Useful resource, well conducted research**

**Rating:** 7
**Confidence:** 4
**Correctness:** Correct
**Clarity:** Clear

**Review:**

See my comments on pros and cons

**Strengths:**

* The proposed benchmark can be very useful for long video understanding. It is a meaningful advancement, as the field urgently needs similar benchmarks so as to drive research progress on long video understanding models.
* The videos are sourced from YouTube, which ensures diversity through searches across various categories. Moreover, the use of manual annotation guarantees high-quality video labels.
* The paper includes a substantial number of experiments, effectively highlighting the performance gap between open-source and closed-source models.

**Additional Feedback:**

No

**Documentation:**

Clearly written

**Ethics:**

No ethical issues are found

**Limitations:**

* Chart Presentations: Some chart presentations require improvement. For example, the font size in Figure 2 is too small, which affects readability.
* Experimental Setup and Conclusions: The primary issue lies in the experimental setup and the conclusions drawn. While using GPT-4 as an evaluation metric is reasonable, the chosen score range of 0-3 diminishes the distinctions between models. Additionally, there are concerns about the experimental setup for some closed-source models. For instance, using only 16 frames for testing long videos (e.g., Gemini and GPT-4) is insufficient, rendering the conclusions from these experiments unreliable. Finally, a more comprehensive evaluation should include some open-source long video models.

[1]MovieChat: From Dense Token to Sparse Memory for Long Video Understanding \
[2]LLaMA-VID: An Image is Worth 2 Tokens in Large Language Models \
[3]MA-LMM: Memory-Augmented Large Multimodal Model for Long-Term Video Understanding \
[4]MiniGPT4-Video: Advancing Multimodal LLMs for Video Understanding with Interleaved Visual-Textual Tokens \
[5]MovieLLM: Enhancing Long Video Understanding with AI-Generated Movies

**Opportunities For Improvement:**

* The authors should provide a more detailed explanation for using this heuristic method to obtain video captions. It is recommended to support this explanation with empirical analysis and additional examples.
* The experimental section should be redesigned and improved by incorporating a wider range of models and evaluation metrics. For further details, please refer to the Limitations section.

**Relation To Prior Work:**

Clearly discussed

**Summary And Contributions:**

This paper introduces a benchmark for long video QA to assess the video understanding capabilities of existing MLLMs across various abilities. The evaluation methods include the “GPT-4” score to better align with human preferences. Extensive experiments demonstrate the effectiveness of the benchmark, and the paper offers key insights into the field of long video understanding.

---

> ### Author Rebuttal · Authors · 2024-08-17
>
> We sincerely thank the reviewer for the thoughtful insights and valuable feedback. We are encouraged to see that the reviewer 8fpf believes our benchmark "very useful for long video understanding" and our work "effectively highlighting the performance gap". Below, we address the main concerns raised by the reviewer.
>
> **Q1. Additional explanation of video captions**
>
> We thank the reviewer for highlighting this issue. It seems there may be a misunderstanding regarding how we obtain video subtitles. We do not employ heuristic methods; instead, we record the video IDs after a thorough cross-check to ensure that the videos are suitable for inclusion in our benchmark. The corresponding subtitles are then retrieved using the **YouTube API** based on these video IDs. The subtitle information obtained is often dense and contains numerous timestamps. Therefore, we combine overlapping segments to ensure that each subtitle represents as complete a sentence as possible. Ultimately, this subtitle information is integrated with questions and visual inputs to facilitate a fine-grained understanding of the video content.
>
> **Q2. Chart Presentations**
>
> We acknowledge the reviewer's constructive criticism regarding the visual presentation of our figures. We appreciate this valuable feedback and will ensure that the font readability in Figure 2 is improved in the updated version, optimizing all visual elements for clarity and accessibility.
>
> **Q3. Experimental Setup and Conclusions**
>
> We greatly appreciate your constructive suggestions. The number of frames is indeed a crucial factor for advanced LVLMs, as it enhances their capabilities. We conducted experiments on both Gemini and GPT-4 with videos sampled at 1 fps, and the results are presented below:
>
> | **Model**          | **overall Mean** | **CP** | **FP-S** | **FP-C** | **HL** | **Perception Mean** | **LR** | **AR** | **RR** | **CSR** | **TR** | **Reasoning Mean** |
> |--------------------|:----------------:|:------:|:--------:|:--------:|:------:|:-------------------:|:------:|:------:|:------:|:-------:|:------:|:-------------------:|
> | GPT-4o-[16f]       | 1.86             | 2.03   | 1.88     | 1.67     | 2.13   | 1.89                | 1.78   | 1.95   | 1.78   | 1.90    | 1.68   | 1.80                |
> | GPT-4o-[1fps]      | 2.15             | 2.23   | 2.24     | 2.01     | 1.9    | 2.19                | 2.11   | 2.12   | 2.17   | 1.94    | 1.97   | 2.08                |
> | Gemini-Pro-v1.5-[16f]   | 1.60        | 1.81   | 1.59     | 1.60     | 2.00   | 1.61                | 1.58   | 1.77   | 1.69   | 1.80    | 1.24   | 1.55                |
> | Gemini-Pro-v1.5-[1fps] | 1.94 | 1.99 | 2.04 | 1.70 | 1.90 | 1.98 | 1.98 | 2.02 | 1.92 | 1.78 | 1.63 | 1.86 |
>
>
> It is evident that an increased number of frames improves model performance. With more video frames, the model's perception ability is enhanced, as neighboring frames can corroborate each other, leading to more accurate perceptions. Moreover, previously missing content due to limited frame sampling is now included, allowing the model to better answer questions that depend on this information, thereby improving its reasoning capability. The only noted decrease in capability is in hallucination, which may arise from the excessive video content misleading the model into generating responses that are not grounded in reality.
>
> Furthermore, we thank reviewer 8fpf for approving our selection of GPT-4 as a judge. We would like to address the selection of a score range of 0-3. This range does not diminish the distinctions between models, as we have established strict and detailed criteria for each score (as denoted in Sec.D in supplementary pdf). Poor-performing models do not receive average scores close to those of advanced models due to the limited number of score segments. The conversion to percentage also reflects the performance differences among the models. For example, the state-of-the-art Video-LLM PLLaVA scored only 30%, while GPT-4o, using the same sampling method, achieved a score of 62%, illustrating the performance gap between the models. Our rating rules are finely defined; further segmentation would be inconvenient and not particularly meaningful.
>
> Regarding more comprehensive evaluations of the additional Video-LLMs mentioned in the comment, we will continuously those models as we progress, as much as possible before the discussion period ends. We promise to add all of the mentioned Video-LLMs in the updated version of this manuscript.
>
> We thank the reviewer once again for their valuable feedback. We hope our detailed responses have adequately addressed your concerns, and we will revise the manuscript to enhance the comprehensiveness of MMBench-Video.

---

> > ### Author Rebuttal · Authors · 2024-08-22
> >
> > **Q3. Experimental Setup and Conclusions (Additional Experiment Results)**
> >
> > Thanks for the comment and we will include all mentioned benchmarks in references, acknowledging their contributions to long Video-LLMs. To enhance the comprehensiveness of our evaluation in MMBench-Video, we included several open-source long video models following the suggestion. We did not include MovieChat due to the significant time costs associated with repeatedly saving and loading video clips. Additionally, MA-LMM requires fine-tuning on the corresponding dataset to improve performance; without this, it tends to perform poorly on zero-shot Video QA benchmarks, making it unsuitable for testing in MMBench-Video.
> >
> > For MiniGPT4-Video, we utilized the Mistral version because of its better performance in zero-shot Video QA tasks. The experimental results are presented in the table below:
> > | **Model**                | **overall Mean** | **CP** | **FP-S** | **FP-C** | **HL** | **Perception Mean** | **LR** | **AR** | **RR** | **CSR** | **TR** | **Reasoning Mean** |
> > |--------------------------|:----------------:|:------:|:--------:|:--------:|:------:|:-------------------:|:------:|:------:|:------:|:-------:|:------:|:------------------:|
> > | InternVL-Chat-v1.5-[8f]  | 1.26             | 1.51   | 1.22     | 1.01     | 1.21   | 1.25                | 0.88   | 1.40   | 1.48   | 1.28    | 1.09   | 1.22               |
> > | PLLaVA-7B                | 1.03             | 1.08   | 1.06     | 0.86     | 0.52   | 1.02                | 0.64   | 1.25   | 1.17   | 0.98    | 1.01   | 1.03               |
> > | LLaMA-VID                | 1.10             | 1.31   | 1.13     | 0.94     | 0.58   | 1.13                | 0.65   | 1.24   | 1.13   | 0.84    | 1.01   | 1.02               |
> > | MiniGPT4-Video (Mistral) | 0.70             | 0.76   | 0.55     | 0.54     | 1.44   | 0.62                | 0.62   | 1.03   | 1.05   | 0.62    | 0.82   | 0.85               |
> > | MovieLLM-Baseline        | 0.78             | 0.93   | 0.74     | 0.63     | 0.08   | 0.75                | 0.46   | 0.93   | 1.05   | 0.59    | 0.90   | 0.84               |
> > | MovieLLM                 | 0.87             | 0.95   | 0.82     | 0.70     | 0.15   | 0.81                | 0.52   | 1.12   | 1.22   | 0.54    | 1.05   | 0.97               |
> >
> > It is evident that LLaMA-VID demonstrates strong performance on MMBench-Video due to its ability to encode multiple frames in long videos, outperforming other short Video-LLMs. However, it still falls short compared to advanced LVLMs, such as InternVL-Chat-v1.5, and exhibits limitations in hallucination capability. MiniGPT4-Video and MovieLLM performed worse than models like Video-LLaVA, which did not undergo long video training and processing techniques. This suggests that training on long video data (over 10 minutes or longer) and employing effective video processing methods may negatively impact the model's understanding of short and medium-length videos. While MovieLLM incorporates synthetic long video data into LLaMA-VID for training, this approach resulted in performance degradation, highlighting the importance of training data quality on model capabilities.

---

> ### Author Response · Authors · 2024-08-21
> **Looking forward to your reply!**
>
> We sincerely appreciate your great efforts in reviewing this paper. Your constructive advice and valuable comments really help improve our paper. Considering the approaching deadline, please, let us know if you have follow-up concerns. We sincerely hope you can consider our reply in your assessment, and we can further address unclear explanations and remaining concerns if any.
>
> Once more, we are appreciated for the time and effort you've dedicated to our paper.

---

> ### Author Response · Authors · 2024-08-30
> **Looking forward to your reply!**
>
> Dear Reviewer 8fpf,
>
> We are writing to express our appreciation for your valuable time and effort in reviewing our paper. We carefully studied your comments and provided a comprehensive response. We will further revise this work according to your suggestion.
>
> To summarize, your three main concerns have been addressed:
>
> 1. **Additional Explanation of Video Captions** : We clarified that the video subtitles are retrieved using the YouTube API and are meticulously processed to ensure they accurately represent the content, facilitating a fine-grained understanding of the videos.
> 2. **Chart Presentations** : We acknowledged the need for better font readability in our figures and committed to optimizing visual elements for enhanced clarity in the updated version.
> 3. **Experimental Setup and Conclusions** : We emphasized that increasing the number of frames improves model performance by providing more accurate perceptions, while also addressing the rationale behind our scoring system and plans for more comprehensive evaluations of additional Video-LLMs.
>
> If you still have any outstanding concerns or if there is anything further you would like to discuss regarding our paper, please do not hesitate to reach out to us by any means. We would be more than willing to address any comments or queries you may have, even after the discussion period.
> Once again, we sincerely appreciate your contribution to the review process. We look forward to any additional feedback you may provide, should you have the opportunity to review our response further.
>
> Thank you!
>
> Sincerely,
>
> MMBench-Video Authors

---

### Official Review · Reviewer_FpG8 · 2024-07-24
**Review of MMBench-Video**

**Rating:** 5
**Confidence:** 4
**Correctness:** Yes.
**Clarity:** Yes.

**Review:**

The paper introduces a novel long-form multi-shot VideoQA benchmark, significantly enhancing existing VideoQA evaluations. Its innovative aspects include using GPT-4 for scoring and a diverse set of question types. However, the evaluation scope is limited to models with no more than 16 frames and lacks tests on larger, more advanced models. Additionally, the paper does not include results from pure LLMs. Overall, while the paper has significant contributions and potential, further improvements and expansions are necessary.

**Strengths:**

- Novel Benchmark: MMBench-Video addresses critical gaps in existing VideoQA benchmarks by including longer videos and a broader range of question types​.
- Balanced Question Distribution: Ensures a more equitable distribution of question types, which better reflects natural human dialogues and provides a more robust evaluation framework.
- Better Scoring Mechanism: Utilizes GPT-4 for scoring, which is a more advanced judge model compared to previous benchmarks that relied on GPT-3.5, leading to more accurate evaluations​.

**Additional Feedback:**

See the above limitations.

**Documentation:**

Yes.

**Ethics:**

No.

**Limitations:**

Yes.

**Opportunities For Improvement:**

- Missing Results: As a video benchmark, it should incorporate the results of pure LLMs, for instance, GPT-4 without videos, to verify if the problems can be solved using only QA (and subtitles).
- Limited Evaluations: The authors only evaluate the MLLMs with no more than 16 frames. Considering that MMBench is a long video benchmark, it should utilize more frames, such as 128, 256, and even frames with 1 FPS. Additionally, the current results only evaluate open-sourced MLLMs with 7B LLMs, which is much behind the closed-sourced MLLMs. Those MLLMs that are larger and more advanced should be evaluated.
- Scoring Costs: What about the scoring costs with GPT-4? Can the powerful open-sourced LLMs like QWen-2 satisfy the scoring tasks?

**Relation To Prior Work:**

Yes.

**Summary And Contributions:**

The paper presents MMBench-Video, a novel long-form multi-shot VideoQA benchmark designed to evaluate the effectiveness of large vision-language models (LVLMs) in understanding video content. This benchmark consists of approximately 600 web videos, providing a diverse set of questions encompassing various interrogative forms. The evaluation framework employs GPT-4 for scoring responses and aims to address existing limitations in VideoQA benchmarks, such as the predominance of short videos and limited question types.

---

> ### Author Rebuttal · Authors · 2024-08-17
>
> We sincerely thank reviewer FpG8 for recognizing the innovation, breadth of question type coverage, and accurate scoring mechanism in our article. That serves as a great encouragement to our work. We have conducted the necessary experiments to address your concerns, and we present our responses below.
>
> **Q1. Missing pure LLM result**
>
> We appreciate the reviewer's insightful comments regarding the necessity of conducting text-only experiments to demonstrate that our benchmark cannot be solved with text information alone. To align with our previous settings, we evaluated GPT-4o on MMBench-Video using only questions and both questions and subtitles, since that GPT-4o is one of the most advanced LVLMs with exceptional capabilities in text reasoning.
>
> | **Model** | **overall Mean** | **CP** | **FP-S** | **FP-C** | **HL** | **Perception Mean** | **LR** | **AR** | **RR** | **CSR** | **TR** | **Reasoning Mean** |
> |----|----|----|----|----|----|----|----|----|----|----|----|----|
> | GPT-4o (question) | 0.25 | 0.11 | 0.1 | 0.11 | 2.29 | 0.19 | 0.13 | 0.47 | 0.42 | 0.09 | 0.4 | 0.34 |
> | GPT-4o (question+vtt) | 0.74 | 0.6 | 0.64 | 0.5 | 2.11 | 0.67 | 0.46 | 0.98 | 0.91 | 0.44 | 0.94 | 0.83 |
> | GPT-4o-[1f] | 0.7 | 0.99 | 0.61 | 0.53 | 2.19 | 0.73 | 0.47 | 0.82 | 0.63 | 0.69 | 0.44 | 0.59 |
> | GPT-4o-[8f] | 1.62 | 1.82 | 1.59 | 1.43 | 1.95 | 1.63 | 1.33 | 1.89 | 1.60 | 1.60 | 1.44 | 1.57 |
>
> The results clearly indicate that relying solely on question inputs, even the most advanced model struggles to address most problems in our dataset. The inclusion of subtitles significantly enhances model performance, yielding a 200% improvement. Subtitles provide detailed information, such as dialogues and video interpretations, enabling the model to gain a comprehensive understanding of the video and answer related questions more effectively. While additional visual input clearly improves model responses, it is noteworthy that using just one frame fails to compete with subtitle input. This is likely because a single frame offers only a partial view of the video, leading to potential mis-reasoning and reduced performance. Nonetheless, the model's perception ability still requires visual input for optimal testing.
>
> **Q2. Limited Evaluations**
>
> We are grateful to the reviewer for highlighting this issue. The number of frames is indeed crucial for advanced LVLMs, as it enhances their capabilities. We conducted experiments on both Gemini and GPT-4 with videos sampled at 1 fps, and the results are summarized below:
>
> | **Model**          | **overall Mean** | **CP** | **FP-S** | **FP-C** | **HL** | **Perception Mean** | **LR** | **AR** | **RR** | **CSR** | **TR** | **Reasoning Mean** |
> |--------------------|:----------------:|:------:|:--------:|:--------:|:------:|:-------------------:|:------:|:------:|:------:|:-------:|:------:|:-------------------:|
> | GPT-4o-[16f]       | 1.86             | 2.03   | 1.88     | 1.67     | 2.13   | 1.89                | 1.78   | 1.95   | 1.78   | 1.90    | 1.68   | 1.80                |
> | GPT-4o-[1fps]      | 2.15             | 2.23   | 2.24     | 2.01     | 1.9    | 2.19                | 2.11   | 2.12   | 2.17   | 1.94    | 1.97   | 2.08                |
> | Gemini-Pro-v1.5-[16f]   | 1.60        | 1.81   | 1.59     | 1.60     | 2.00   | 1.61                | 1.58   | 1.77   | 1.69   | 1.80    | 1.24   | 1.55                |
> | Gemini-Pro-v1.5-[1fps] | 1.94 | 1.99 | 2.04 | 1.70 | 1.90 | 1.98 | 1.98 | 2.02 | 1.92 | 1.78 | 1.63 | 1.86 |
>
> It is evident that an increased number of frames improves model performance. With more video frames, the model's perception ability is enhanced, as neighboring frames can corroborate each other, leading to more accurate perceptions. Moreover, previously missing content due to limited frame sampling is now included, allowing the model to better answer questions that depend on this information, thereby improving its reasoning capability. The only noted decrease in capability is in hallucination, which may arise from the excessive video content misleading the model into generating responses that are not grounded in reality.
>
> Furthermore, the models evaluated in our work lag behind the current advancements in LVLM capabilities and sizes. Given that we have conducted experiments on the most advanced closed-source LVLMs, including Gemini-Pro-v1.5 and GPT-4o, we have also incorporated two larger and more powerful open-source LVLMs to explore the upper limits of advanced open-source models. The evaluation results of two below models and additional large-scale LVLMs and Video-LLMs will be included in the updated version of this manuscript.
>
> | **Model** | **overall Mean** | **CP** | **FP-S** | **FP-C** | **HL** | **Perception Mean** | **LR** | **AR** | **RR** | **CSR** | **TR** | **Reasoning Mean** |
> |----|------------------|--------|----------|----------|--------|---------------------|--------|--------|--------|---------|--------|--------------------|
> | InternVL2-26B-[16f] | 1.41 | 1.56 | 1.48     | 1.23     | 0.52   | 1.42                | 1.06   | 1.61   | 1.45   | 1.38    | 1.23   | 1.35               |
> | LLaVA-NeXT-Video-34B-HF-[32f] | 1.14 | 1.35   | 1.15     | 0.97     | 0.58   | 1.14                | 0.64   | 1.38   | 1.3    | 1.27    | 1.03   | 1.13               |
>
> As observed, the current large open-source LVLMs still exhibit a significant gap compared to GPT-4o under the same sampling conditions. The InternVL series performs better compared to models of similar volume. However, both models struggle with hallucination capabilities, likely due to most LVLMs not developing robust refusal mechanisms when faced with inadequate input.

---

> > ### Author Rebuttal · Authors · 2024-08-17
> >
> > **Q3. Scoring Costs**
> >
> > We are very grateful to the reviewer for the innovative suggestions. The scoring cost for GPT-4 is \$5 - $7 per evaluation, making it advantageous to explore powerful open-source LLMs for rating. We conducted a series of experiments using Qwen2-72B-Instruct as a judge with predictions from Video-LLaVA and GPT-4o. The overall scoring and mean absolute error (MAE) of the judges, compared to human preference results, are summarized below:
> >
> > | **Judge Model** | **Video-LLaVA (Avg. Score)** | **GPT-4o (Avg. Score)** |
> > |---------------------------|:---------------:|:-----------:|
> > | GPT-3.5-Turbo (1106)      | 2.09            | 2.45        |
> > | GPT-3.5-Turbo (0613)      | 1.8             | 2.11        |
> > | GPT-4-Turbo (1106)        | 1.05            | 1.62        |
> > | GPT-4-Turbo (0125)        | 0.9             | 1.61        |
> > | Qwen2-72B-Instruct        | 1.15            | 1.8         |
> >
> > | **Judge Model**      | **Video-LLaVA (MAE w. Human)** | **GPT-4o (MAE w. Human)** |
> > | -------------------- | :----------------------------: | :-----------------------: |
> > | GPT-3.5-Turbo (1106) |              0.98              |           0.815           |
> > | GPT-3.5-Turbo (0613) |              0.89              |           0.685           |
> > | GPT-4-Turbo (1106)   |              0.36              |           0.295           |
> > | GPT-4-Turbo (0125)   |              0.36              |           0.255           |
> > | Qwen2-72B-Instruct   |              0.41              |           0.32            |
> >
> > The results suggest that GPT-4-Turbo is the model most consistent with human ratings, while Qwen2 also demonstrates stable and reliable scoring abilities compared to GPT-3.5-Turbo. The MAE for Qwen2-72B is significantly lower than that of GPT-3.5-Turbo and only about 10% higher than the rating model we used (GPT-4-Turbo-1106) on average. The results indicate the objectivity and fairness of GPT-4 ratings, which do not favor its own predictions, while also suggesting the feasibility of using a powerful open-source LLM for ratings.
> >
> > We thank the reviewer once again for their valuable feedback. We hope our detailed responses have addressed your concerns, and we will revise the manuscript to enhance the comprehensiveness of MMBench-Video.

---

> ### Author Response · Authors · 2024-08-21
> **Looking forward to your reply!**
>
> We sincerely appreciate your great efforts in reviewing this paper. Your constructive advice and valuable comments really help improve our paper. Considering the approaching deadline, please, let us know if you have follow-up concerns. We sincerely hope you can consider our reply in your assessment, and we can further address unclear explanations and remaining concerns if any.
>
> Once more, we are appreciated for the time and effort you've dedicated to our paper.

---

> ### Author Response · Authors · 2024-08-30
> **Looking forward to your reply!**
>
> Dear Reviewer FpG8,
>
> We are writing to express our appreciation for your valuable time and effort in reviewing our paper. We carefully studied your comments and provided a comprehensive response. We will further revise this work according to your suggestion.
>
> To summarize, your three main concerns have been addressed:
>
> 1. **Missing pure LLM result:** We conducted text-only experiments with GPT-4o, demonstrating that our benchmark cannot be effectively solved using text information alone, as even with subtitles, visual input significantly enhances model performance.
> 2. **Limited evaluations:** We expanded our evaluations by including both larger open-source and proprietary LVLMs, confirming that an increased number of frames improves model performance, while also addressing the limitations in current LVLM capabilities.
> 3. **Scoring costs:** We compared various judge models, concluding that GPT-4-Turbo is most consistent with human ratings and that using a powerful open-source LLM like Qwen2-72B for scoring is feasible and cost-effective.
>
> If you still have any outstanding concerns or if there is anything further you would like to discuss regarding our paper, please do not hesitate to reach out to us by any means. We would be more than willing to address any comments or queries you may have, even after the discussion period.
> Once again, we sincerely appreciate your contribution to the review process. We look forward to any additional feedback you may provide, should you have the opportunity to review our response further.
>
> Thank you!
>
> Sincerely,
>
> MMBench-Video Authors

---

### Official Review · Reviewer_kFa6 · 2024-07-25
**VideoQA with More Diverse Domain and Longer Video Context, but Lacks Relevant Experiment Results and Dataset Collection Details and quality Control.**

**Rating:** 5
**Confidence:** 4

**Review:**

Overall, a new free-form video QA dataset that expands the domain coverage and fine-grained categories of QAs. However, the dataset design lacks significant novelty, and the evaluation setup and dataset control are weakly justified. Despite focusing on multi-shot and lengthy videos, the experiments do not thoroughly investigate their impact on model performance. Additionally, the evaluation method lacks a comparison with MC setups like MVBench, and no human evaluation is conducted to establish oracle performance or validate the setup.

**Pros**
- Focuses on multi-shot, long videos, distinguishing it from existing benchmarks that typically cover short, trimmed clips.
- Includes more videos in the wild from Youtube that cover more diverse video domains.
- More fine-grained, hierarchical categories of questions are introduced to the dataset to better assess performance across various domains and capabilities.

**Cons**
- No significant innovations in dataset curation or quality control. It seems to be extension of other Video QA benchmarks, except applied to more diverse and longer videos.
- Missing analysis on number of shots to the video models and only plays around with 8 or 16 frames setting. Limited number of frames is considered as evaluation with max number of frames as 16, despite the emphasis on multi-shot and long videos. Considering there are 32.5 number of shots in the video, the number of frames at inference is limited to achieve the best performance in the task.
    - In addition, breakdown of the performance by video time length and number of shots should be included to determine if the model lacks reasoning over long videos
- Missing details of how annotations are curated and questions are constructed.
   - How are the question types determined and controlled? How is the dataset verified? What template and instructions were annotators given to formulate the free form questions and answers? How to ensure the quality of the dataset and restrict the answers? How are the volunteers collected?
- No human evaluation results are reported on the dataset to determine the oracle performance and validating the dataset.
- Missing comprehensive qualitative examples of each question type.
- Missing justification on using LLM-based evaluation over multiple choice, as LLMs can potentially be unreliable and the authors fail to validate their evaluation capabilities. Why should models evaluate on this benchmark instead of MC-based eval such as MVBench, which has well-defined answer and evaluation design?
- Comparisons with image VQA Benchmarks in Table 4 are not directly relevant to the video-based challenges of MMVideo-Bench. More appropriate set up would be to compare with MVBench and how the model performance varies across the benchmark.
- The dataset's scale is relatively small, with only 609 video clips and 2000 QAs, despite the diverse domains considered in the dataset.

**Strengths:**

- Expands the scope of video domains and introduces a more detailed, hierarchical categorization of questions from existing VideoQA dataset.
- See the pros in the Review.

**Additional Feedback:**

None

**Clarity:**

The paper could benefit from improved clarity in its language and presentation. It should include more qualitative examples for each question type to provide a clearer understanding of the types of videos and questions it covers, so that readers have better overall view of what kind of videos and QAs it covers, similar to MVBench.

**Correctness:**

Yes, the claims seem correct. The evaluation and experiment design could benefit by incorporating different shots and more number of frames.

**Documentation:**

Data collection and organization are not well documented, and dataset control is not performed.

**Ethics:**

Potentially there might be inappropriate videos included in the dataset, which may need to be filtered out in collection stage.

**Limitations:**

Limitations are addressed but more frames of videos should be incorporated for evaluation to achieve the best performance in proposed long, multi-shot videos.

**Opportunities For Improvement:**

- Various experiment setups for analyzing effect of number of shots and video length should be considered.
- More details of dataset control and annotation process should be included.
- Qualitative examples per question type should be included.
- Human evaluation numbers should be reported.
- See the cons in the Review.

**Relation To Prior Work:**

The paper does not compare directly to MVBench.

**Summary And Contributions:**

MMBench Video, unlike other VideoQA benchmark, incorporates lengthy videos from Youtube to create QA dataset targeted for temporal reasoning with a diverse range of video topics and fine-grained capabilities. The questions are formulated based on the introduced 3-level hierarchical taxonomy with 26 leafs inherited from MMBench.  Similar to VideoQA benchmarks, GPT-4 is used to automatically assess the free form answers scaled from 0-3. Authors compare the temporal indispensability of existing benchmark by measuring  the performance difference of GPT4o when provided 1 frame vs 8 frames. The difference is prominent for MMBench video dataset compared to existing VideoQA benchmarks. The authors evaluate on open-source video LLMs, image LVLMs, and proprietary LVLMs. Out of the open sourced models, the image based LVLMS InternVL-chat gets the best performance, while gpt4-o achieves the best overall when increasing number of frames from 1, 8 to 16. Incorporating speech information from video subtitles also further boosts the gpt4-o performance.

---

> ### Author Rebuttal · Authors · 2024-08-17
>
> We sincerely thank the reviewer for the thoughtful insights and valuable feedback. We are encouraged to see that the reviewer kFa6 believes our work "focuses on multi-shot, long videos" and features "more fine-grained, hierarchical categories of questions." Below, we address the main concerns raised by the reviewer.
>
> **Q1. Clarification of innovation and the construction of our dataset**
>
> We would emphasize that our dataset is not an extension of other Video QA benchmarks, but rather "**a comprehensive benchmark for assessing LVLMs' proficiency in video understanding**" as noted by reviewer aXTw. Our innovations regarding dataset curation and quality control can be summarized as follows:
>
> 1. **Dataset Curation:** Previous Video QA benchmarks often rely on the transformation of existing datasets (MSRVTT -> MSRVTT-QA, etc.). This results in limitations concerning data richness and the semantic enrichment of QA pairs. In those benchmarks, QA pairs are automatically modified from the original data samples, even suffering from syntactic issues. MVBench is also sourced from existing video datasets, making it prone to overfitting on corresponding training datasets. Unlike previous benchmarks, MMBench-Video is a human-collected and annotated dataset featuring long-form, diverse videos sourced from the web, covering a wide array of topics. After identifying the competencies necessary for video comprehension, we instructed human annotators to find corresponding YouTube videos and formulate questions based on the content. A second annotator performs cross-checking to ensure the quality of the questions. This comprehensive curation process guarantees the reliability and thoroughness of our dataset.
>
> 2. **Quality control:** We first implement cross-checking to enhance data quality. Questions irrelevant to the video content or overly simplistic are filtered out during quality check, and misrepresented or inaccurate answers are corrected. We further employ a quality control method using proprietary LVLM. For each VideoQA, GPT-4v samples a single random frame as visual input and conducts inference four times with distinct random seeds. We then use GPT-4 to evaluate the responses and compute the average score for each question. Questions with an average score of 2.5 or higher across the four responses are excluded. This approach allowed us to remove 246 temporally dispensable questions from the dataset (More details in supp Sec. C.1).
>
> Moreover, compared to large-scale eval benchmarks,  a smaller dataset allows for fast assessments of model performance while reducing resource consumption. This is advantageous for rapidly model iteration and capability optimization. Our dataset encompasses 26 leaf capabilities covering cognitive aspects of video comprehension, 16 major video categories, and various video lengths. The high-quality human-annotated question-answer pairs enhance the value of our dataset, which is a critical aspect of the evaluation benchmark.
>
> **Q2. Impact of Increased Frame Number**
>
> We appreciate the reviewer's and valuable suggestions. The number of frames is a crucial factor for LVLMs' task performance. Most existing Video-LLMs sample 8 / 16 frames from videos for training and evaluation (e.g., Video-LLaVA, PLLaVA), thus we adopt it as our default evaluation setting. Moreover, we conducted additional 1-fps experiments on proprietary LVLMs to investigate the performance improvements.
>
> | **Model** | **overall Mean** | **CP** | **FP-S** | **FP-C** | **HL** | **Perception Mean** | **LR** | **AR** | **RR** | **CSR** | **TR** | **Reasoning Mean** |
> |--|:--:|:--:|:--:|:--:|:--:|:--:|:--:|:--:|:--:|:--:|:--:|:--:|
> | GPT-4o-[16f] | 1.86 | 2.03 | 1.88 | 1.67 | 2.13 | 1.89 | 1.78 | 1.95 | 1.78 | 1.90 | 1.68 | 1.80 |
> | GPT-4o-[1fps] | 2.15 | 2.23 | 2.24 | 2.01 | 1.9 | 2.19 | 2.11 | 2.12 | 2.17 | 1.94 | 1.97 | 2.08 |
> | Gemini-Pro-v1.5-[16f] | 1.60 | 1.81 | 1.59 | 1.60 | 2.00 | 1.61 | 1.58 | 1.77 | 1.69 | 1.80 | 1.24 | 1.55 |
> | Gemini-Pro-v1.5-[1fps] | 1.94 | 1.99 | 2.04 | 1.70 | 1.90 | 1.98 | 1.98 | 2.02 | 1.92 | 1.78 | 1.63 | 1.86 |
>
> It is evident that an increased number of frames improves model performance. With more video frames, the model's perception ability is enhanced, as neighboring frames can corroborate each other, leading to more accurate perceptions. Moreover, previously missing content due to limited frame sampling is now included, allowing the model to better answer questions that depend on this information, thereby improving its reasoning capability. The only noted decrease in capability is in hallucination, which may arise from the excessive video content misleading the model into generating responses that are not grounded in reality.
>
> **Q3. Analysis performance by video time length**
>
> We are grateful to the reviewer for their innovative suggestions regarding our experimental section. The length of the video and the number of shots are indeed key factors affecting model performance. Videos with fewer and shorter shots may perform better at lower frame counts, while longer or multi-shot videos require more visual content to fully leverage the model's capabilities.
>
> Fig. 1 in our rebuttal pdf illustrates the trend of model scores in relation to shot count and video length. It is evident that the performance of GPT-4o, sampled at different frame rates, declines as video length increases, whereas the performance of open-source models such as InternVL-Chat-v1.5 and Video-LLaVA remains relatively stable. Compared to video length, model performance is more significantly influenced by the number of video shots. With over 50 shots in a video, the performance of GPT-4o drops to 75% of its original score. This suggests that the model's performance is more closely tied to the number of shots than to video length, as the frequent shot transitions make it more challenging for the model to comprehend the video, resulting in lower scores.

---

> > ### Author Rebuttal · Authors · 2024-08-17
> >
> > **Q4. Human Evaluation of result**
> >
> > We express our gratitude to the reviewer for raising this question. Some aspects of the human evaluation results included in our work are presented in Table 7. It is clear that GPT-4 aligns better with human preferences and shows a lower mean absolute error compared to human ratings. Thus, it is reasonable and effective to select GPT-4 as the judging model for MMBench-Video. A larger scale of human evaluation is unnecessary due to GPT-4's high alignment with human judgments, and the time and labor costs associated with large-scale tagging are considerable. Furthermore, almost all existing video benchmarks, such as MVBench (MCQ) and VideoChatGPT-benchmark (open-ended), also do not provide human evaluation results for their entire datasets. Human judgment can falter in complex contexts requiring specialized knowledge. Therefore, it is not practical or necessary to report overall human evaluation results.
> >
> > **Q5. Qualitative examples of each question type**
> >
> > We are immensely grateful to the reviewer for highlighting these important issues. Some qualitative examples are provided in our supplementary material, specifically Figures 8 to 10. However, we acknowledge that these examples do not cover all question types, which impacts the overall presentation of our work and may not provide the reviewer with sufficient examples to assess the benchmark thoroughly. We commit to completing these qualitative examples in the appendix in future versions, and we thank reviewer kFa6 for the reminder.
> >
> > **Q6. Comparison between MVBench and MMBench-Video**
> >
> > We greatly appreciate the reviewer for their insightful comments regarding the comparison between MVBench and MMBench-Video. While MVBench is a valuable MC-based evaluation video benchmark, it does have some shortcomings:
> >
> > 1. **Video Source:** The videos in MVBench were collected from 11 public datasets, and the tasks are restricted to specific situations or scenes that are rarely encountered in real life (e.g., the CLEVER dataset). This limitation does not accurately reflect model performance in everyday scenarios. Additionally, existing public datasets allow some models to fine-tune on their training sets, potentially inflating scores under MVBench evaluation. In contrast, our videos are sourced from YouTube based on topics, making them more relevant to daily viewing habits and better revealing the model's ability to comprehend common video content without the risk of directional fine-tuning. The measures discussed in Q1 further ensure data quality.
> >
> > 2. **Evaluation Bias:** Although MC-based evaluation offers certain advantages in terms of cost and objectivity, it also presents biases and issues. Some Video-LLMs exhibit significant positional bias when responding to MCQ questions, often favoring options presented earlier in the list (e.g., option A). This can lead to underperforming models appearing to perform well on the MCQ benchmark. Furthermore, existing benchmarks often rely on rigid patterns or regularization rules to extract options in model predictions, which can result in poor ratings for models that do not conform to these patterns. In contrast, open-ended questions align better with the data format used for model fine-tuning, avoiding option-ordering bias and providing a more natural assessment of model performance. The advanced evaluation judge, GPT-4, with its more reasonable scoring and higher alignment with human ratings, further demonstrates reliability compared to other LLMs and human evaluators.
> >
> > The comparison with image VQA benchmarks in Table 4 illustrates that existing Video-LLMs struggle with challenging questions at the image level, complicating the resolution of temporal understanding issues unique to the video domain. We believe that a necessary step in addressing video-based challenges is first to tackle perception and reasoning regarding static images. Compared to other benchmarks (e.g., the MSRVTT dataset), MMBench-Video reveals a greater performance gap between models, thereby providing a clearer picture of improvements in model performance.
> >
> > We thank the reviewer once again for their valuable feedback. We hope our detailed responses have addressed the reviewer's concerns, and we will revise the manuscript to enhance the comprehensiveness of MMBench-Video.

---

> ### Author Response · Authors · 2024-08-21
> **Looking forward to your reply!**
>
> We sincerely appreciate your great efforts in reviewing this paper. Your constructive advice and valuable comments really help improve our paper. Considering the approaching deadline, please, let us know if you have follow-up concerns. We sincerely hope you can consider our reply in your assessment, and we can further address unclear explanations and remaining concerns if any.
>
> Once more, we are appreciated for the time and effort you've dedicated to our paper.

---

> ### Author Response · Authors · 2024-08-30
> **Looking forward to your reply!**
>
> Dear Reviewer kFa6,
>
> We are writing to express our appreciation for your valuable time and effort in reviewing our paper. We carefully studied your comments and provided a comprehensive response. We will further revise this work according to your suggestion.
>
> To summarize, your six main concerns have been addressed:
>
> 1. **Clarification of innovation and dataset construction:** We emphasized that our dataset, MMBench-Video, is uniquely human-collected and annotated, focusing on long-form videos with diverse content, and not merely an extension of existing benchmarks. Our rigorous quality control ensures its reliability and thoroughness.
> 2. **Impact of increased frame number:** We acknowledged the importance of frame count, demonstrating that a higher number of frames improves the model's perception and reasoning capabilities, despite a slight increase in hallucinations.
> 3. **Analysis of performance by video time length:** We agreed that video length and shot count are crucial factors, with our findings showing that shot count has a more significant impact on model performance than video length, particularly for models like GPT-4o.
> 4. **Human evaluation of results:** We clarified that GPT-4's high alignment with human judgments justifies its selection as the judging model for MMBench-Video, making large-scale human evaluations unnecessary due to their high cost and limited added value.
> 5. **Qualitative examples of each question type:** We acknowledged the lack of comprehensive qualitative examples in the current version and committed to expanding these in the appendix for future versions, ensuring a more thorough evaluation of the benchmark.
> 6. **Comparison between MVBench and MMBench-Video:** We highlighted the advantages of MMBench-Video over MVBench, such as the use of more realistic video content sourced from YouTube and a more reliable open-ended evaluation approach, which reduces biases inherent in MC-based evaluations.
>
> If you still have any outstanding concerns or if there is anything further you would like to discuss regarding our paper, please do not hesitate to reach out to us by any means. We would be more than willing to address any comments or queries you may have, even after the discussion period.
> Once again, we sincerely appreciate your contribution to the review process. We look forward to any additional feedback you may provide, should you have the opportunity to review our response further.
>
> Thank you!
>
> Sincerely,
>
> MMBench-Video Authors

---

### Official Review · Reviewer_aXTw · 2024-07-25
**A new benchmark to evaluate LVLMs in understanding video content**

**Rating:** 6
**Confidence:** 4
**Correctness:** Yes
**Clarity:** Yes

**Review:**

Pros:
1. The dataset is well-collected and annotated. All answers are manually annotated which preserves the quality of the dataset.
2. The taxonomy of questions and videos is well-defined.
3. The paper is well-structured. The evaluations of many models and the open-source contribution can benefit the community.

Cons:
1. The evaluation is open-ended and model-based, which may be easy to hack and may contain biases.
2. The size of the dataset is relatively small.
3. While the concept of temporal indispensability is interesting, the compared datasets mostly focus on only short videos and are appearance-based. It is better to compare this metric with other datasets such as EgoSchema, VideoMME.

**Strengths:**

1. The dataset is well-collected and annotated. All answers are manually annotated which preserves the quality of the dataset.
2. The taxonomy of questions and videos is well-defined.
3. The proposed concept of temporal indispensability is interesting.
4. The paper is well-structured. The evaluations of many models and the open-source contribution can benefit the community.

**Additional Feedback:**

N/A

**Documentation:**

Yes

**Ethics:**

I have not checked the videos in the dataset, there is a risk that the videos in the dataset contain ethical violations. It is the author's duty to ensure the videos comply the ethical requirements.

**Limitations:**

Yes

**Opportunities For Improvement:**

1. The author may consider to also include some multiple choice questions as a reference.
2. The author may consider adding new videos to the dataset.
3. The author should consider comparing the temporal Indispensability with other datasets like EgoSchema and Video-MME, Next-GQA, etc. The current comparison is not convincing.

**Relation To Prior Work:**

Yes

**Summary And Contributions:**

The paper aims to provide a comprehensive benchmark for assessing LVLMs' proficiency in video understanding, which traditional VideoQA benchmarks fail to cover adequately. The proposed MMBench-Video benchmark includes long videos from YouTube and free-form questions to mirror practical use cases. It includes a detailed ability taxonomy to test models' temporal reasoning skills. All the question answers are human-annotated, which is very valuable. It uses approximately 600 web videos covering 16 major categories, reflecting common video topics in daily life. Each video ranges from 30 seconds to 6 minutes to evaluate models on (a little) longer video content. The benchmark includes around 2,000 original QA pairs. Since the question answer is open-ended, the authors use GPT-4 for automated assessment.

---

> ### Author Rebuttal · Authors · 2024-08-16
>
> We thank the reviewer for the encouraging comments and address the main concerns below. We are pleased to note that the reviewer considers our dataset to be "well-collected and annotated" and believes our work "provides a comprehensive benchmark for assessing LVLMs' proficiency in video understanding." These points are precisely what we aim to highlight.
>
> **Q1. Bias Analysis of Open-ended Evaluation**
>
> We appreciate the reviewer for raising this important concern. Model-based evaluation is the most effective approach for reasonably assessing the differences between model outputs and ground truth answers, as it accounts for semantic similarity and assigns higher scores to predictions that, while differing in length, are similar in meaning.
>
> As illustrated in Fig. 6 and Table 6, 7, we selected GPT-4 for evaluation due to its reasonable scoring and low mean absolute error compared to human judgment, which helps minimize bias.
>
> 1. **Easy to Hack**: It is easier to manipulate the model when it simply rates its own predictions. However, our method involves scoring after a thorough comparison between ground truth and predictions under strict rating principles, making it more accurate and resistant to manipulation. Moreover, both LVLMs and Video-LLMs produce low-risk outputs in everyday visual scenes. We also tested several methods to induce GPT-4 to give higher ratings, but all were unsuccessful.
>
> 2. **Potential Biases**: We conducted an experiment to assess the use of a powerful open-source LLM for rating. The results indicate that the GPT series does not tend to give higher scores to its own predictions, demonstrating fairness in scoring based on the consistency between predictions and ground truth. This further justifies our choice of GPT-4 as the evaluator for the MMBench-Video benchmark.
>
> | **Judge Model** | **Video-LLaVA** | **GPT-4o** |
> |--|:--:|:--:|
> | GPT-3.5-Turbo (1106) | 2.09 | 2.45 |
> | GPT-3.5-Turbo (0613) | 1.8 | 2.11 |
> | GPT-4-Turbo (1106) | 1.05 | 1.62 |
> | GPT-4-Turbo (0125) | 0.9 | 1.61 |
> | Qwen2-72B-Instruct | 1.15 | 1.8 |
>
> **Q2. Size of dataset**
>
> Compared to larger evaluation benchmarks, a smaller dataset has the advantage of facilitating quicker model performance assessments while minimizing resource consumption. This is beneficial for rapid model iteration and dedicated performance optimizations. Moreover, for evaluation purposes, the benchmark quality always overwhelms the data quantity. MMBench-Video contains high-quality human-annotated question-answer pairs that cover 26 fine-grained capabilities and 16 major video categories with various video lengths. The extensive coverage and data quality demonstrate that our dataset is both concise and effective.
>
> **Q3. Comparison of temporal indispensability**
>
> We appreciate the reviewer's insightful observation regarding temporal indispensability in relation to other datasets. It is important to note that current temporal indispensability experiments primarily focus on traditional VideoQA datasets due to their widespread use. Many video-LLMs rely on these datasets to showcase their performance. However, the short durations, fewer shots, single contexts, and simplistic responses associated with these datasets are no longer suitable for assessing model performance. As a newly proposed VideoQA benchmark, MMBench-Video features a variety of video lengths and shot counts, demonstrating greater temporal indispensability compared to conventional VideoQA benchmarks in our experiments.
>
> Additionally, for other mentioned benchmarks (EgoSchema, Next-GQA, etc.), it's also essential to evaluate their temporal indispensability in comparison to MMBench-Video. We will release these results later during the discussion period, as these datasets contain substantial volumes of data.
>
> **Q4. MCQ Reference**
>
> Thanks for the sensible suggestion to "include some multiple choice questions as a reference", which we will take into account and may include some multiple choice questions later as appropriate.
>
> However, there are also 2 problems in the construction and assessment process of multiple-choice questions:
>
> 1. **Construction:**  It is challenging to create effective yet hard distractor options when formulating similar problems. Powerful models can achieve significant performance with only the question and options provided, making it difficult to differentiate the capabilities of different models. For instance, simply choosing `A` can yield approximately 30% accuracy in MVBench, while advanced Video-LLMs like VideoChat2 only achieves 50%. In contrast, SOTA LLMs like GPT-4o struggles to answer correctly with only the question input, revealing a substantial gap between question-only input and inputs with subtitles or frames under MMBench-Video.
> 2. **Assessment:** Research indicates that apparent emergent abilities can arise under nonlinear or discontinuous metrics [1]. This suggests that multiple-choice questions may reflect unrealistic sudden changes in capability due to minor model optimizations (such as response format), whereas open-ended questions provide a smoother representation of changes in the model's capabilities and are more realistic.
>
> Overall, open-ended questions are more reliable for evaluating the model's true capabilities. Meanwhile, we will consider including some multiple-choice questions in the total question set, and perform the corresponding ablation study.
>
> **Q5. Ethic Concerns**
>
> All videos in our benchmark are human-selected based on appropriate value propositions and undergo a second manual quality check to ensure there are no ethical violations. Thank you for highlighting this important aspect!
>
> We thank the reviewer once again for the valuable feedback. We hope our detailed responses have addressed the reviewer's concerns, and we will revise the manuscript to enhance the comprehensiveness of MMBench-Video.
>
> [1] Schaeffer R, Miranda B, Koyejo S. Are emergent abilities of large language models a mirage? NeurIPS, 2024.

---

> > ### Author Rebuttal · Authors · 2024-08-19
> >
> > **Q3. Comparison of temporal indispensability (Experiment Results)**
> >
> > We conducted a temporal indispensability experiment using the EgoSchema, Next-GQA and MMBench-Video datasets. GPT-4o was used for prediction under 1-frame and 8-frame settings, while GPT-4-turbo was employed for rating. Given that Video-MME divides its video duration into three parts, and considering that videos longer than 15 minutes are challenging to evaluate quickly and fall outside the scope of our dataset, we excluded long-duration videos to better align with MMBench-Video. The experiment results are as follows:
> > | Benchmark             | EgoSchema |      | Next-GQA |       | Video-MME (short+medium) |      | MMBench-Video |      |
> > |-----------------------|-----------|------|----------|-------|--------------------------|------|---------------|------|
> > | Input Frames          | 1         | 8    | 1        | 8     | 1                        | 8    | 1             | 8    |
> > | Original Score        | 0.65      | 0.70 | 0.775    | 0.838 | 0.54                     | 0.68 | 0.78          | 1.63 |
> > | Normalized Score      | 65        | 70   | 77.5     | 83.8  | 54                       | 68   | 26            | 54.3 |
> > | Score-[1f]/score-[8f] | 92.86%    |      | 92.5%    |       | 79.4%                    |      | 47.80%        |      |
> >
> > The results indicate that MMBench-Video exhibits significant temporal indispensability compared to recent datasets such as EgoSchema, Next-GQA, and Video-MME. The ratio exceeding 90% for EgoSchema and Next-GQA suggests that the questions in these benchmarks can be effectively answered from a single static image, implying they do not sufficiently challenge the model's temporal reasoning abilities. Although Video-MME showed slightly better performance in this regard, it still falls short of fully assessing the model's temporal capabilities. In contrast, the MMBench-Video dataset, with all videos under six minutes long and a ratio of less than 50%, provides a robust evaluation of the model's ability to understand video content across time domains.

---

> > > ### Comment · Reviewer_aXTw · 2024-08-27
> > > **Thanks for the rebuttal**
> > >
> > > I have follow-up concerns on the temporal indispensability. Since other benchmarks are multiple choice QAs and only the proposed is open ended, I am not sure if this temporal indispensability measure is correct to compare MCQ benchmarks with this open-ended benchmark.

---

> > > > ### Author Rebuttal · Authors · 2024-08-28
> > > >
> > > > We appreciate Reviewer aXTw for bringing up this concern. Firstly, it's important to clarify that our paper did compare the temporal indispensability with open-ended VideoQA datasets (such as MSRVTT and ActivityNet in the original paper), highlighting the temporal significance of MMBench-Video. The additional benchmarks mentioned by the reviewer aXTw are mostly MCQ benchmarks, so we follow their original setting for evaluating the temporal indispensability.

---

> > > > > ### Comment · Reviewer_aXTw · 2024-08-28
> > > > >
> > > > > Since ActivityNet-QA and MSRVTT-QA primarily focus on short videos, and it is widely recognized that using fewer frames can yield reasonable performance on these datasets, I believe this comparison does not address my concerns. It is hard to compare the temporal indispensability metric with existing datasets that are designed for longer video understanding. I understand that resolving this issue is not straightforward given the current circumstances. However, I recommend that the authors give more attention to this metric, as the current comparison is not fair, and achieving a truly fair comparison is quite difficult.

---

> > > > > > ### Author Rebuttal · Authors · 2024-08-29
> > > > > >
> > > > > > Thanks to reviewer aXTw for your quick reply. Since the past open-ended video datasets mostly use short videos, they are inherently characterized by poor temporal indispensability. We continue to believe that the guided-response nature of the MCQ's options is insufficient to mask the nature of its poor ability to examine models in terms of temporal indispensability. We will consider subsequently transforming recent MCQ benchmarks (e.g., Video-MME, Next-GQA) into open-ended questions to further test their temporal indispensability.

---

> > ### Author Response · Authors · 2024-08-30
> >
> > Dear Reviewer aXTw,
> >
> > Thank you once again for your valuable comments and the time you've taken to review our work. We hope our responses have satisfactorily addressed your questions.
> >
> > As the discussion period approaches its end, we wanted to check if is there anything further that you would like us to clarify or elaborate on.
> >
> > We would be more than happy to provide any further clarifications, discussions, or additional experiments to ensure all your concerns are fully addressed.
> >
> > Sincerely,
> >
> > MMBench-Video Authors

---

> ### Author Response · Authors · 2024-08-21
> **Looking forward to your reply!**
>
> We sincerely appreciate your great efforts in reviewing this paper. Your constructive advice and valuable comments really help improve our paper. Considering the approaching deadline, please, let us know if you have follow-up concerns. We sincerely hope you can consider our reply in your assessment, and we can further address unclear explanations and remaining concerns if any.
>
> Once more, we are appreciated for the time and effort you've dedicated to our paper.

---

### Official Review · Reviewer_AJQp · 2024-07-27

**Rating:** 7
**Confidence:** 5
**Correctness:** Yes
**Clarity:** Yes

**Review:**

The paper is well-written, The proposed benchmark is comprehensive and addresses some of the limitations of existing VideoQA benchmarks. It is an overall good contribution to the research community, and some improvements to be made includes adding a few additional experiments and baselines.

Please see Strengths and Opportunities For Improvement for pros and cons of this work. I am willing to increase my scores if the cons are properly addressed.

**Strengths:**

- The proposed benchmark is a valuable contribution to the field, as it addresses some of the limitations of existing VideoQA benchmarks. It measures various aspects of LVLMs in both perception and reasoning.
- The GPT-4 evaluation scheme improves the accuracy and robustness of VideoQA evaluations. The Score-[1f] / Score-[8f] results are interesting and useful for determining the temporal indispensability of a benchmark.
- The paper includes a comprehensive evaluation of several state-of-the-art LVLMs on the proposed benchmark.

**Additional Feedback:**

N/A

**Documentation:**

Yes

**Limitations:**

Yes

**Opportunities For Improvement:**

- Figure quality could be improved. For instance, the color choice in figure 1 makes it difficult to distinguish between InternVL and Gemini. The font choice and its size in figure 2 also makes it not easy to read.
- I am wondering how much the performance can be improved by further increasing the number of frames. For example, both Gemini-Pro-1.5 and GPT-4o support an extensive number of frames, which makes sampling at 1FPS or ½ FPS [1] likely possible on videos from the proposed benchmark. Could you further increase the number of frames and see how this can affect the model performances on MMBench-Video?
- VILA-1.5 [2] is also a strong open-source LVLMs that can process both images and videos. Please include it in the benchmark.

[1] https://video-mme.github.io/home_page.html
[2] VILA: On Pre-training for Visual Language Models. CVPR 2024. https://github.com/NVlabs/VILA

**Relation To Prior Work:**

Yes

**Summary And Contributions:**

This paper introduces MMBench-Video, a new benchmark for evaluating large vision-language models (LVLMs) on video understanding tasks. The benchmark includes a diverse set of long-form videos from YouTube, along with human-annotated questions and answers. The authors also introduce an enhanced evaluation methodology using GPT-4 for scoring, aiming to provide more accurate and robust assessments.

---

> ### Author Rebuttal · Authors · 2024-08-16
>
> We appreciate the thoughtful feedback from the reviewer and are encouraged by the recognition of MMBench-Video's strengths in addressing the limitations of existing VideoQA benchmarks and enhancing the evaluation approach with GPT-4 for scoring. We address the reviewer's concerns below:
>
> **Q1. Figure Quality**
>
> We acknowledge the reviewer's suggestion regarding the visual presentation of our figures and express our gratitude for this valuable feedback. The color contrast in Figure 1 and font readability in Figure 2 is duly noted. We will rectify these shortcomings in the update version, ensuring the visual elements are optimized for clarity and accessibility.
>
> **Q2. Impact of Increased Frame Number**
>
>
> We appreciate the reviewer's insights and valuable suggestions. The number of frames is a critical factor for advanced LVLMs, as it extends their potential capabilities. We conducted experiments on Gemini-Pro-1.5 and GPT-4o with videos sampled at 1 fps, and the results are presented below:
>
> | **Model**          | **overall Mean** | **CP** | **FP-S** | **FP-C** | **HL** | **Perception Mean** | **LR** | **AR** | **RR** | **CSR** | **TR** | **Reasoning Mean** |
> |--------------------|:----------------:|:------:|:--------:|:--------:|:------:|:-------------------:|:------:|:------:|:------:|:-------:|:------:|:-------------------:|
> | GPT-4o-[16f]       | 1.86             | 2.03   | 1.88     | 1.67     | 2.13   | 1.89                | 1.78   | 1.95   | 1.78   | 1.90    | 1.68   | 1.80                |
> | GPT-4o-[1fps]      | 2.15             | 2.23   | 2.24     | 2.01     | 1.9    | 2.19                | 2.11   | 2.12   | 2.17   | 1.94    | 1.97   | 2.08                |
> | Gemini-Pro-v1.5-[16f]   | 1.60        | 1.81   | 1.59     | 1.60     | 2.00   | 1.61                | 1.58   | 1.77   | 1.69   | 1.80    | 1.24   | 1.55                |
> | Gemini-Pro-v1.5-[1fps] | 1.94 | 1.99 | 2.04 | 1.70 | 1.90 | 1.98 | 1.98 | 2.02 | 1.92 | 1.78 | 1.63 | 1.86 |
>
> It is evident that an increased number of frames improves model performance. With more video frames, the model's perception ability is enhanced, as neighboring frames can corroborate each other, leading to more accurate perceptions. Moreover, previously missing content due to limited frame sampling is now included, allowing the model to better answer questions that depend on this information, thereby improving its reasoning capability. The only noted decrease in capability is in hallucination, which may arise from the excessive video content misleading the model into generating responses that are not grounded in reality.
>
> **Q3. VILA performance on MMBench-Video**
>
> We thank the reviewer for highlighting this powerful LVLM. We have incorporated this model into our benchmark, and the experimental results are as follows:
>
> | **Model**               | **overall Mean** | **CP** | **FP-S** | **FP-C** | **HL** | **Perception Mean** | **LR** | **AR** | **RR** | **CSR** | **TR** | **Reasoning Mean** |
> |-------------------------|:----------------:|:------:|:--------:|:--------:|:------:|:-------------------:|:------:|:------:|:------:|:-------:|:------:|:-------------------:|
> | InternVL-Chat-v1.5-[8f] | 1.26             | 1.51   | 1.22     | 1.01     | 1.21   | 1.25                | 0.88   | 1.40   | 1.48   | 1.28    | 1.09   | 1.22                |
> | VILA1.5-13B-[16f] | 1.37             | 1.5    | 1.46     | 1.3      | 0.26   | 1.4                 | 0.81   | 1.56   | 1.27   | 1.42    | 1.29   | 1.29                |
> | VILA1.5-40B-[16f]      | 1.63             | 1.79   | 1.74     | 1.37     | 0.47   | 1.65                | 1.14   | 1.8    | 1.63   | 1.49    | 1.46   | 1.54                |
> | Gemini-Pro-v1.5-[16f]   | 1.60             | 1.81   | 1.59     | 1.60     | 2.00   | 1.61                | 1.58   | 1.77   | 1.69   | 1.80    | 1.24   | 1.55                |
> | GPT-4o-[16f]            | 1.86             | 2.03   | 1.88     | 1.67     | 2.13   | 1.89                | 1.78   | 1.95   | 1.78   | 1.90    | 1.68   | 1.80                |
>
> For VILA, we uniformly sample 16 frames from the video as inputs. We evaluate two variants of VILA 1.5: 13B and 40B. The results show that VILA is an outstanding LVLM, achieving better performance across most capabilities compared to InternVL-Chat-v1.5. Notably, VILA1.5-40B displays impressive performance on MMBench-Video, demonstrating comparable performance to Gemini-Pro v1.5 across most capabilities. However, VILA exhibits weakness in hallucination relative to its other strong capabilities, which may be attributed to its weaker refusal to answer.
>
> We thank the reviewer once again for their valuable feedback. We hope our detailed responses have addressed the reviewer's concerns, and we will revise the manuscript to enhance the comprehensiveness of MMBench-Video.

---

> ### Author Response · Authors · 2024-08-21
> **Looking forward to your reply!**
>
> We sincerely appreciate your great efforts in reviewing this paper. Your constructive advice and valuable comments really help improve our paper. Considering the approaching deadline, please, let us know if you have follow-up concerns. We sincerely hope you can consider our reply in your assessment, and we can further address unclear explanations and remaining concerns if any.
>
> Once more, we are appreciated for the time and effort you've dedicated to our paper.

---

> > ### Comment · Reviewer_AJQp · 2024-08-26
> > **Response on rebuttal**
> >
> > Thank you for the comprehensive response. Most of my concerns have been addressed, but I still have a follow-up question on some details of the experiment: I am wondering how VILA1.5-40B-[16f] setting was tested? It seems that VILA1.5-40B represents each frame with 256 tokens, which leads to 4096 tokens in total for 16 frames. While the context window of its LLM is also 4096 tokens, making it not able to fit in 16 frames and instructions. A common setting to test VILA1.5-40B is using 14 frames (e.g. [Video-MME](https://video-mme.github.io/home_page.html#leaderboard) and [MLVU](https://github.com/JUNJIE99/MLVU?tab=readme-ov-file#trophy-mini-leaderboard)), and putting 16 frames could probably result in low-quality outputs. Could the authors provide more details and an explanation on this point?

---

> > > ### Author Rebuttal · Authors · 2024-08-28
> > >
> > > Many thanks to reviewer AJQp for the follow-up response! To ensure a fair comparison with other Video-LLMs mentioned in our paper, we tested VILA with a 16-frame input. To deal with issues related to context windows, we first calculate the total token length of the input before performing the forward. Once we find the token length exceeds the context window size, we progressively merging input frames with horizontal concatenation until the input fits the context window size. For example, if the token length exceed, we first merge 1st and 2nd frames, and then 3rd and 4th frames, and so on. Typically, we only need to perform the merging op 1 / 2 times to fit VILA context window.
> > >
> > > Additionally, we know that adaptively merging frames may not be a standard practice for Video-LLMs, so we further conduct experiments with 14 frames on VILA, and present the results below:
> > >
> > > | **Model**               | **overall Mean** | **CP** | **FP-S** | **FP-C** | **HL** | **Perception Mean** | **LR** | **AR** | **RR** | **CSR** | **TR** | **Reasoning Mean** |
> > > |-------------------------|:----------------:|:------:|:--------:|:--------:|:------:|:-------------------:|:------:|:------:|:------:|:-------:|:------:|:-------------------:|
> > > | InternVL-Chat-v1.5-[8f] | 1.26             | 1.51   | 1.22     | 1.01     | 1.21   | 1.25                | 0.88   | 1.40   | 1.48   | 1.28    | 1.09   | 1.22                |
> > > | VILA1.5-13B-[14f]       | 1.36             | 1.51   | 1.45     | 1.26     | 0.24   | 1.39                | 0.80   | 1.52   | 1.30   | 1.40    | 1.28   | 1.28                |
> > > | VILA1.5-40B-[14f]       | 1.61             | 1.78   | 1.72     | 1.35     | 0.47   | 1.63                | 1.12   | 1.78   | 1.61   | 1.48    | 1.45   | 1.52                |
> > > | VILA1.5-13B-[16f]       | 1.37             | 1.5    | 1.46     | 1.3      | 0.26   | 1.4                 | 0.81   | 1.56   | 1.27   | 1.42    | 1.29   | 1.29                |
> > > | VILA1.5-40B-[16f]       | 1.63             | 1.79   | 1.74     | 1.37     | 0.47   | 1.65                | 1.14   | 1.8    | 1.63   | 1.49    | 1.46   | 1.54                |
> > > | Gemini-Pro-v1.5-[16f]   | 1.60             | 1.81   | 1.59     | 1.60     | 2.00   | 1.61                | 1.58   | 1.77   | 1.69   | 1.80    | 1.24   | 1.55                |
> > > | GPT-4o-[16f]            | 1.86             | 2.03   | 1.88     | 1.67     | 2.13   | 1.89                | 1.78   | 1.95   | 1.78   | 1.90    | 1.68   | 1.80                |
> > >
> > > The results indicate that VILA performs similarly with both 14 and 16-frame inputs. This consistency suggests that increasing the number of input frames may have reached the upper limit of the model's performance, and additional frames do not significantly enhance the model's capabilities.

---

> > ### Author Response · Authors · 2024-08-30
> >
> > Dear Reviewer AJQp,
> >
> > Thank you once again for your valuable comments and the time you've taken to review our work. We hope our responses have satisfactorily addressed your questions.
> >
> > As the discussion period approaches its end, we wanted to check if is there anything further that you would like us to clarify or elaborate on.
> >
> > We would be more than happy to provide any further clarifications, discussions, or additional experiments to ensure all your concerns are fully addressed.
> >
> > Sincerely,
> >
> > MMBench-Video Authors

---

> > > ### Comment · Reviewer_AJQp · 2024-09-01
> > > **Response**
> > >
> > > Thank you for the detailed response. I have increased the rating. Please include the experimental results (1 fps and VILA) in the next updated version of the paper.

---

> > > > ### Author Response · Authors · 2024-09-02
> > > >
> > > > Thank you again for taking the time to review our work and for recognizing our response, the comments you made were precious. We promise to add the results of our experiments (1 fps and VILA) to our paper.

---

### Author Response · Authors · 2024-09-01

Dear Area Chair,

Thank you for your dedicated efforts in organizing the review process for our paper. We sincerely appreciate the time and feedback provided by all reviewers.

Common concerns raised by the reviewers that have been addressed during the discussion period:

1.**Dataset and Innovation:** MMBench-Video is a uniquely human-collected and annotated dataset focusing on diverse, long-form videos, ensuring reliability and reducing biases compared to MVBench.

2.**Frame Number Impact:** Increasing frame numbers enhance model perception and reasoning, although it may slightly increase hallucinations; visual input significantly boosts performance over text-only input.

3.**Human Evaluation and Scoring Costs:** GPT-4 aligns well with human judgments, reducing the need for costly human evaluations, and using open-source LLMs like Qwen2-72B for scoring is cost-effective.

4.**Experimental Setup and Temporal Indispensability:** MMBench-Video effectively challenges models' temporal understanding, with models like LLaMA-VID performing well but still falling short of advanced LVLMs like InternVL-Chat-v1.5.

Current Reviewing status:

**AJQp:** Maintains acceptance post-discussion and is willing to increase scores.

**aXTw:** Maintains acceptance post-discussion and has slight concerns over temporal indispensability.

**kFa6, FpG8:** have itemized their concerns and look forward to further replies.

**8fpf:** have itemized his concerns and look forward to further replies, rating currently stays as acceptance.

We value reviewers' feedback and remain open to addressing concerns. Their constructive advice and valuable comments really helped improve our paper. As we approach the deadline, we eagerly anticipate the response from the remaining reviewers. We truly appreciate your support in facilitating productive discussions throughout this process.

Thanks for your time and support.

Sincerely,

MMBench-Video Authors

---

### Decision · Program_Chairs · 2024-09-26

**Decision:**

Accept (Poster)

**Comment:**

This paper presents a benchmark dataset for evaluating vision-language models in video understanding that incorporates long videos with free-form questions. Questions are human-annotated and GPT-4 is used for evaluation.
The dataset is a clear contribution to the community, and after the rebuttal and discussion the paper received two accepts, marginally above accept, and two marginally below accept. The reviewer concerns address the dataset size and frequency which the authors address in the rebuttal. The reviewers agrree that the dataset is useful for fine grained benchmarking of long video understanding.